# OODROBUSTBENCH: BENCHMARKING AND ANALYZING ADVERSARIAL ROBUSTNESS UNDER DISTRIBUTION SHIFT

## ABSTRACT

Existing works have made great progress in improving adversarial robustness, but typically test their method only on data from the same distribution as the training data, i.e. in-distribution (ID) testing. As a result, it is unclear how such robustness generalizes under input distribution shifts, i.e. out-of-distribution (OOD) testing. This is a concerning omission as such distribution shifts are unavoidable when methods are deployed in the wild. To address this issue we propose a benchmark named OODRobustBench to comprehensively assess OOD adversarial robustness using 23 dataset-wise shifts (i.e. naturalistic shifts in input distribution) and 6 threat-wise shifts (i.e., unforeseen adversarial threat models). OODRobustBench is used to assess 706 robust models using 60.7K adversarial evaluations. This large-scale analysis shows that: 1) adversarial robustness suffers from a severe OOD generalization issue; 2) ID robustness correlates strongly with OOD robustness, in a positive linear way, under many distribution shifts. The latter enables the prediction of OOD robustness from ID robustness. Based on this, we are able to predict the upper limit of OOD robustness for existing robust training schemes. The results suggest that achieving OOD robustness requires designing novel methods beyond the conventional ones. Last, we discover that extra data, data augmentation, advanced model architectures and particular regularization approaches can improve OOD robustness. Noticeably, the discovered training schemes, compared to the baseline, exhibit dramatically higher robustness under threat shift while keeping high ID robustness, demonstrating new promising solutions for robustness against both multi-attack and unforeseen attacks.

## 1 INTRODUCTION

Adversarial attack poses a serious threat to real-world machine learning models, and various approaches have been developed to defend against such attacks. Previous work (Athalye et al., 2018) has shown that adversarial evaluation is critical to the study of adversarial robustness since an unreliable evaluation can often give a false sense of robustness. However, we believe that even state-of-the-art evaluation benchmarks (like RobustBench (Croce et al., 2021)) suffer from a severe limitation: they only consider ID generalization where test data comes from the same distribution as the training data. Since distribution shifts are inevitable in the real world, it is crucial to assess how adversarial robustness is affected when the test distribution differs from the training one.

Although OOD generalization is extensively studied for clean accuracy with rich conclusions (Hendrycks & Dietterich, 2019; Taori et al., 2020; Miller et al., 2021; Baek et al., 2022; Zhao et al., 2022; Yang et al., 2022), there is little known about the OOD generalization of adversarial robustness. To fill this blank, this paper presents for the first time, a comprehensive benchmark, **OODRobustBench**, for assessing out-of-distribution adversarial robustness. Our benchmark is analogous and complementary to RobustBench which is used for assessing in-distribution adversarial robustness. OODRobustBench includes two categories of distribution shift: dataset shift and threat shift (see Fig. 1). Dataset shift denotes test data that has different characteristics from the training data due to varying conditions under which the samples are collected: for images, these include but are not limited to corruptions, background, and viewpoint. OODRobustBench contains 23 such dataset shifts and assesses adversarial robustness to such data using the attack seen by the

model during training. Threat shift denotes a variation between training and test adversarial threat models. In other words, threat shift assesses a model's robustness to unseen adversarial attacks applied to ID test data. OODRobustBench employs six different types of threat shifts. Adversarial robustness is evaluated for each type of shift to comprehensively assess OOD robustness.

With OODRobustBench, we analyze the OOD generalization behavior of 706 well-trained robust models (a total of 60.7K adversarial evaluations). This model zoo covers a diversity of architectures, robust training methods, data augmentation techniques and training set-ups to ensure the conclusions drawn from this assessment are general and comprehensive. This large-scale analysis reveals that:

- **Adversarial robustness suffers from a severe OOD generalization issue**. Robustness degrades on average by 18%/31%/24% under distribution shifts for CIFAR10 $\ell_\infty$, CIFAR10 $\ell_2$ and ImageNet $\ell_\infty$ respectively. Among individual models, the degradation is non-uniform: the higher the ID robustness of the model, the more robustness degrades under distribution shift. Furthermore, an abnormal catastrophic drop in robustness under noise shifts is observed in some methods, e.g., under Gaussian noise shift, the robustness of Rade & Moosavi-Dezfooli (2022) drops by 46% whereas the average drop is just 9%.

- **ID and OOD accuracy/robustness correlate strongly in a linear trend under many shifts** (visualized in Fig. 1). This enables the prediction of OOD performance from ID performance. Furthermore, contrary to the previous finding on standardly-trained (ST) models (Miller et al., 2021), adversarially-trained (AT) models exhibit a much stronger linear correlation between ID and OOD accuracy under most corruption shifts on CIFAR10.

- **Simply following the existing methodology may be insufficient to achieve OOD robustness**. Based on the discovered linear trend, we predict that even the best achievable OOD performance is still far below 100%. For instance, the predicted upper bound of robustness on ImageNet $\ell_\infty$ is only 43% under the dataset shifts.

Overall, this work reveals that most existing robust models including the state-of-the-art ones are vulnerable to distribution shifts and demonstrates that the existing approaches to improve ID robustness may be insufficient to achieve high OOD robustness. To ensure safe deployment in the wild, we advocate for the assessment of OOD robustness in future models and for the development of new approaches that can cope with distribution shifts better and achieve OOD robustness beyond our prediction.

## 2 RELATED WORK

**Robustness under dataset shift**. Early work (Sehwag et al., 2019) studied the generalization of robustness to novel classes that are unseen during training, e.g., the class of digit "7" in MNIST is novel to the classifier training on CIFAR10. On the other hand, our setup only considers the input distribution shift and not the unforeseen classes. Recently, Sun et al. (2022b) studied the OOD generalization of certified robustness under corruption shifts for a few state-of-the-art methods. In contrast, we focus on empirical robustness instead of certified robustness. Alhamoud et al. (2023) is the most relevant work. They studied the generalization of robustness from multiple source domains to an unseen domain. Different from them, the models we examine are trained on only one source domain, which is the most common set-up in the existing works of adversarial training (Croce et al., 2021). Moreover, we also cover much more diverse distribution shifts, models and training methods than Sun et al. (2022b) and Alhamoud et al. (2023) so that the conclusion drawn in this work is more general and comprehensive. Except for a few exceptions (Geirhos et al., 2020; Sun et al., 2022a; Rusak et al., 2020; Ford et al., 2019), previous work on generalization to input distribution shifts has not considered adversarial robustness. Hence, work on robustness to OOD data and adversarial attacks has generally happened in parallel, as exemplified by RobustBench (Croce et al., 2021) which provides independent benchmarks for assessing performance on corrupt data and adversarial threats.

**Robustness against unforeseen adversarial threat models**. It was observed that naive adversarial training (Madry et al., 2018) with only one single $\ell_p$ threat model generalizes poorly to unforeseen $\ell_p$ threat models, e.g., higher perturbation bound (Stutz et al., 2020), different $p$-norm (Tramer & Boneh, 2019; Maini et al., 2020; Croce & Hein, 2022), or non-$\ell_p$ threat models including color transformation ReColor (Laidlaw & Feizi, 2019), spatial transformation StAdv (Xiao et al., 2018), LPIPS-bounded attacks PPGD and LPA (Laidlaw et al., 2021) and many others (Kaufmann et al.,

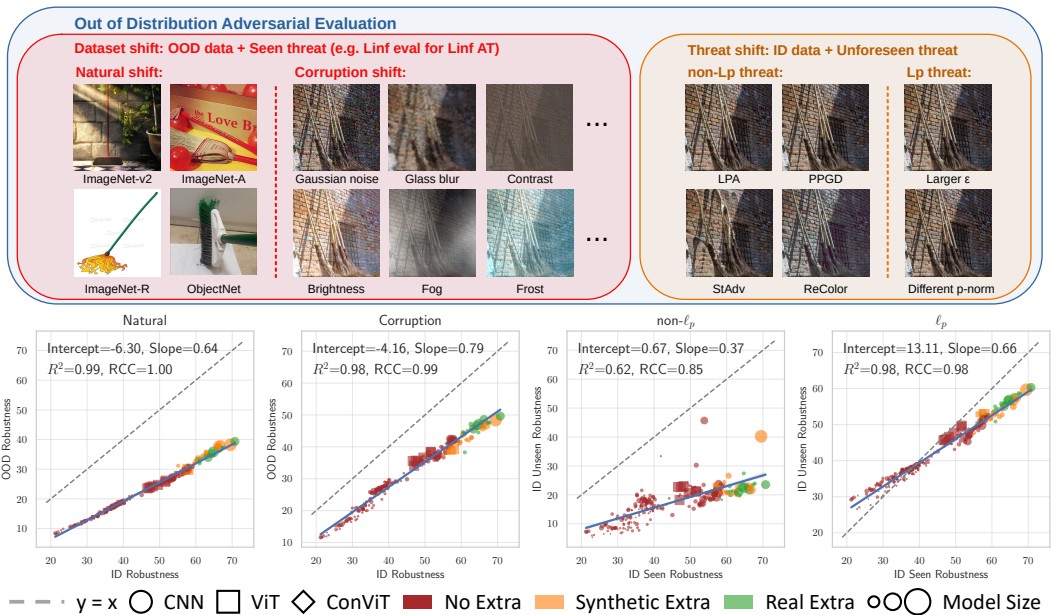

Figure 1: The construction of OODRobustBench (top) and the correlation between ID and OOD robustness under 4 types of distribution shift for CIFAR10 $\ell_\infty$ (bottom). Each marker represents a model and is annotated by its training set-up. The solid blue line is the fitted linear correlation. The dashed gray line ($y = x$) represents perfect generalization where OOD robustness equals ID robustness. Deviation from the dashed line indicates robustness degradation under the respective distribution shift.

2023). A line of works (Tramer & Boneh, 2019; Maini et al., 2020) defends against a union of $\ell_p$ threat models by training with multiple $\ell_p$ threat models jointly, which makes these threat models no longer unforeseen. PAT (Laidlaw et al., 2021) replaces $\ell_p$ bound with LPIPS (**?**) in adversarial training and achieves high robustness against several unforeseen attacks. Alternatively, Dai et al. (2022) proposes variation regularization in addition to $\ell_p$ adversarial training and improves unforeseen robustness. We complement the existing works by conducting a large-scale analysis on the unforeseen robustness of $\ell_p$ robust models trained by varied methods and training set-ups. We are thus able to provide new insights into the generalization of robustness to unforeseen threat models and identify effective yet previously unknown approaches to enhance unforeseen robustness.

## 3  BENCHMARKING OUT-OF-DISTRIBUTION ADVERSARIAL ROBUSTNESS

This section first proposes the benchmark OODRobustBench and then presents the benchmark results for state-of-the-art robust models.

### 3.1  OODROBUSTBENCH

OODRobustBench is designed to simulate the possible data distribution shifts that might occur in the wild and evaluate adversarial robustness in the face of them. It focuses on two types of distribution shifts: dataset shift and threat shift. *Dataset shift*, $\text{OOD}_d$, denotes the distributional difference between training and test raw datasets. *Threat shift*, $\text{OOD}_t$, denotes the difference between training and evaluation *threat models*, a special type of distribution shift. The original test set drawn from the same distribution as the training set is considered ID. The variant dataset with the same classes yet where the distribution of the inputs differs is considered OOD.

**Dataset shift**. To represent diverse data distribution in the wild, OODRobustBench includes multiple types of dataset shifts from two sources: *natural* and *corruption*. For natural shifts, we adopt four different variant datasets per source dataset: CIFAR10.1 (Recht et al., 2018), CIFAR10.2 (Lu et al., 2020), CINIC (Darlow et al., 2018), and CIFAR10-R (Hendrycks et al., 2021a) for CIFAR10, and

ImageNet-v2 (Recht et al., 2019), ImageNet-A (Hendrycks et al., 2021b), ImageNet-R (Hendrycks et al., 2021a), and ObjectNet (Barbu et al., 2019) for ImageNet. For corruption shifts, we adopt, from the corruption benchmarks (Hendrycks & Dietterich, 2019), 15 types of common corruption in four categories: Noise (gaussian, impulse, shot), Blur (motion, defocus, glass, zoom), Weather (fog, snow, frost) and Digital (brightness, contrast, elastic, pixelate, JPEG). Each corruption has five levels of severity. Overall, the dataset-shift testbed consists of 79 ($4 + 15 \times 5$) subsets. App. B.1 describes the details of the above datasets and data processing.

Accuracy and robustness are evaluated on the ID dataset and every OOD dataset. To compute the overall performance of $OOD_d$, we first average the result of natural and corruption shifts:

$$R_c(f) = \mathbb{E}_{i \in \{\text{corruptions}\}, j \in \{\text{severity}\}} R_{i,j}(f) \tag{1}$$

$$R_n(f) = \mathbb{E}_{i \in \{\text{naturals}\}} R_i(f) \tag{2}$$

where $R(\cdot)$ returns accuracy or adversarial robustness and $f$ denotes the model to be assessed. Next, we average the above two results to get the overall performance of the dataset shift as

$$R_{ood}(f) = (R_c(f) + R_n(f))/2 \tag{3}$$

**Threat shift**. For $\ell_p$ AT models, OODRobustBench consists of six unforeseen attacks to simulate threat shifts as in Laidlaw et al. (2021). They are categorized into two groups, non-$\ell_p$ and $\ell_p$, according to whether they are bounded by the $\ell_p$ norm or not. $\ell_p$ shift includes attacks with the same $p$-norm but larger $\epsilon$ and with different $p$-norm. Non-$\ell_p$ shift includes the attacks PPGD, LPA, ReColor, and StAdv. Specifically, the perturbation bound is 0.5 for PPGD, 0.5 for LPA, 0.05 for StAdv and 0.06 for ReColor. The number of iterations is 40 for PPGD and LPA regardless of dataset, is 100 for StAdv and ReColor on CIFAR10 and 200 on ImageNet. The overall robustness under threat shift, $OOD_t$, is simply the mean of these six unforeseen attacks. App. B.2 describes the specific parameters of these attacks.

### 3.2 OOD Performance and Ranking

The benchmark results for CIFAR10 $\ell_\infty$, $\ell_2$ and ImageNet $\ell_\infty$ are in Tabs. 1, 2 and 3 respectively.

**Robustness degrades significantly under distribution shift**. For models trained to be robust for CIFAR10 $\ell_\infty$ (Fig. 2), CIFAR10 $\ell_2$ (Fig. 7) and ImageNet $\ell_\infty$ (Fig. 8), the average drop in robustness (ID adversarial accuracy - OOD adversarial accuracy) is 18%/20%/27% under dataset shift and 18%/42%/22% under threat shift.

Robustness degradation varies greatly across different kinds of shifts. Robustness severely degrades for a subset of shifts: whereas the average robustness degradation of $OOD_d$ is 18% on CIFAR10 $\ell_\infty$, some shifts like CIFAR10-R, fog and contrast degrade by 38%, 30% and 32%, respectively.

Robustness degradation also varies greatly across individual models. As shown in Fig. 2, the distribution of robustness degradation for most shifts spreads over a wide range, suggesting a large variation across individual models. Specifically, we find that the higher the ID robustness of the model, the more robustness degrades under the shifts. For example, the top method in Tab. 1 degrades by 30% of robustness, while the bottom method degrades by only 18%. This suggests that while the great progress has been made on improving ID robustness, we only gain diminishing returns under the distribution shifts.

Catastrophic degradation under noise shifts. We highlight that some methods suffer from abnormally catastrophic degradation in robustness under noise shifts constituting the outliers under noise shifts in Fig. 2. This issue is most severe on Rade & Moosavi-Dezfooli (2022) whose robustness falls by 43%/46%/38% under impulse/Gaussian/shot noise, whereas the average drop is 12%/9%/8% (discussed in App. E). A similar yet milder drop is also observed on Debenedetti et al. (2023) and models trained with some advanced data augmentations like AutoAugment (Cubuk et al., 2019).

**Higher ID robustness generally implies higher OOD robustness but not always**. The ranking of ID and OOD robustness is generally consistent (see the last two columns of Tabs. 1, 2 and 3). However, some of the models do break this trend. In Tab. 1, the ranking of Rade & Moosavi-Dezfooli (2022) drops from 22 to 57 due to catastrophic degradation, while the ranking of Pang et al. (2020) jumps from 70 to 3 due to its superior robustness under threat shift (analyzed in Sec. 5.3).

Table 1: Performance, evaluated with OODRobustBench, of state-of-the-art models trained on CI-FAR10 to be robust to $\ell_\infty$ attacks and corruptions. Top 3 results under each metric are highlighted by **bold** and/or underscore. Severe ranking discrepancies are marked in red. The column "OOD" gives the overall OOD robustness which is the mean of the robustness to $OOD_d$ and $OOD_t$.

| Training | Method | Accuracy (%) | | Robustness (%) | | | | Ranking (Rob.) | |
|---|---|---|---|---|---|---|---|---|---|
| Threat | | ID | $OOD_d$ | ID | $OOD_d$ | $OOD_t$ | OOD | ID | OOD |
| $\ell_\infty$ | Wang et al. (2023) (WRN-70-16) | 93.25 | 76.04 | **70.76** | **44.49** | 35.80 | **40.14** | 1 | 2 |
| | Bai et al. (2023) | **95.23** | 79.09 | 69.50 | **43.32** | **46.71** | **45.01** | 2 | 1 |
| | Cui et al. (2023) | 92.16 | 74.88 | 67.77 | 42.48 | 35.48 | 38.98 | 3 | 4 |
| | Wang et al. (2023) (WRN28-10) | 92.44 | 75.04 | 67.34 | 42.34 | 35.26 | 38.80 | 4 | 5 |
| | Rebuffi et al. (2021) | 92.23 | 74.89 | 66.79 | 42.60 | 33.65 | 38.12 | 5 | 9 |
| | Gowal et al. (2021b) | 88.74 | 70.68 | 66.24 | 42.76 | 33.65 | 38.20 | 6 | 8 |
| | Gowal et al. (2021a) | 91.10 | 73.24 | 66.03 | 42.58 | 34.00 | 38.29 | 7 | 7 |
| | Huang et al. (2023) | 91.58 | 73.89 | 65.87 | 41.70 | 33.34 | 37.52 | 8 | 12 |
| | Rebuffi et al. (2021) | 88.50 | 70.65 | 64.82 | 41.43 | 33.90 | 37.66 | 9 | 10 |
| | Xu et al. (2022) | 93.69 | 77.22 | 64.74 | 39.67 | 37.02 | 38.34 | 10 | 6 |
| | Sehwag et al. (2022) | 87.21 | 69.20 | 62.72 | 40.77 | 32.35 | 36.56 | 17 | 15 |
| | Rade & Moosavi-Dezfooli (2022) | 88.16 | 69.45 | 60.98 | 35.10 | 30.20 | 32.65 | 22 | 57 |
| | Wu et al. (2020) | 88.25 | 69.82 | 60.14 | 38.20 | 31.39 | 34.80 | 26 | 27 |
| | Carmon et al. (2019) | 89.69 | 71.57 | 59.80 | 36.74 | 31.12 | 33.93 | 28 | 38 |
| | Wang et al. (2020) | 87.50 | 70.23 | 56.75 | 35.50 | 32.68 | 34.09 | 52 | 35 |
| | Pang et al. (2020) | 85.14 | 66.96 | 53.84 | 32.45 | **46.20** | 39.33 | 70 | 3 |
| | Zhang et al. (2020) | 84.52 | 65.94 | 53.65 | 32.95 | 31.84 | 32.40 | 71 | 59 |
| | Rice et al. (2020) | 85.34 | 66.46 | 53.52 | 32.07 | 27.89 | 29.98 | 72 | 89 |
| | Zhang et al. (2019) | 84.92 | 66.51 | 52.68 | 31.68 | 26.54 | 29.11 | 76 | 99 |
| | Wong et al. (2020) | 83.34 | 64.96 | 43.33 | 25.35 | 24.82 | 25.08 | 111 | 112 |
| Corruption | Diffenderfer et al. (2021) | **96.56** | 83.54 | 1.00 | 0.50 | 0.00 | 0.25 | 261 | 261 |
| | Kireev et al. (2022) | 94.75 | **80.39** | 0.16 | 0.06 | 0.00 | 0.03 | 262 | 262 |

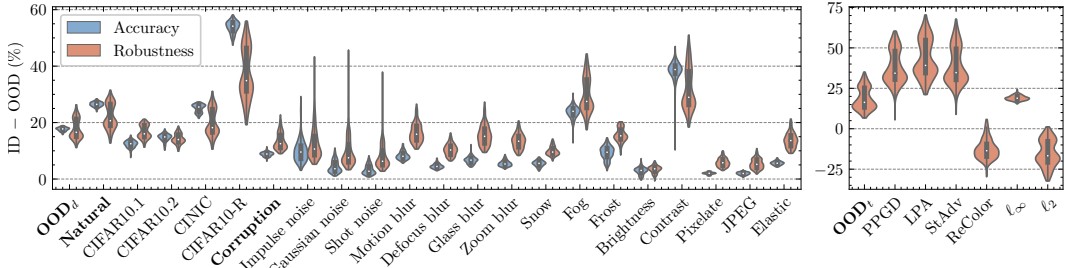

Figure 2: Degradation of accuracy and robustness under various distribution shifts for CIFAR10 $\ell_\infty$.

# 4 THE LINEAR TREND BETWEEN ID AND OOD PERFORMANCE

In Sec. 3.2, we have seen that the robustness ranking among the models is generally preserved under the distribution shift. Here, we further show that the ID and OOD adversarial accuracy of existing models are strongly and linearly correlated under many shifts. We also find that adversarial training, compared to standard training, significantly improves the linear correlation between ID and OOD accuracy under most corruption shifts. Based on the discovered linear trend, we predict the upper limit of OOD performance for the existing robust training techniques and observe a large gap between the predicted upper limit and the ideal best OOD robustness, suggesting that novel approaches beyond the existing ones are required to achieve OOD robustness.

The following result is based on a large-scale analysis including over 60K OOD evaluations of 706 models. 187 of these models were retrieved from RobustBench or other published works so as to include current state-of-the-art methods, and the remaining models were trained by ourselves. These models are mainly trained in three set-ups: CIFAR10 $\ell_\infty$, CIFAR10 $\ell_2$ and ImageNet $\ell_\infty$. They cover a wide range of model architectures, model sizes, data augmentation methods, training and regularization techniques. More detail is given in App. C.2.

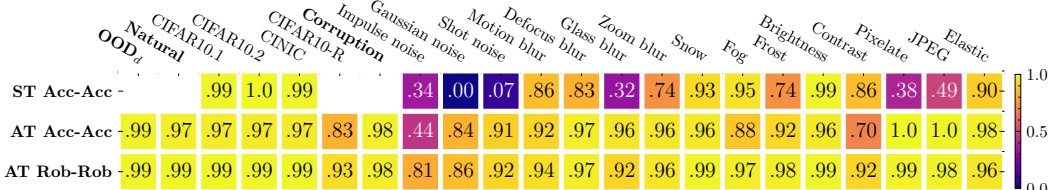

Figure 3: $R^2$ of regression between ID and OOD performance for Standardly-Trained (ST) and Adversarially-Trained (AT) models under various dataset shifts for CIFAR10 $\ell_\infty$. Higher $R^2$ implies stronger linear correlation. The results for ST models were copied from Miller et al. (2021). Some results of ST are missing (blank cells) because they were not reported in Miller et al. (2021).

## 4.1 DATASET SHIFT

This section studies how ID and OOD accuracy/robustness correlate under dataset shifts. We fit a linear regression on four pairs of metrics (Acc-Acc, Rob-Rob, Acc-Rob, and Rob-Acc) for each dataset shift and each training setup (CIFAR10 $\ell_\infty$, CIFAR10 $\ell_2$ and ImageNet $\ell_\infty$). Taking Acc-Rob as an example, a linear model is fitted with ID accuracy as the observed variable $x$ and OOD adversarial robustness as the target variable $y$. The result of regression for each shift is given in App. I. Below are the major findings.

**ID accuracy (resp. robustness) strongly correlates with OOD accuracy (resp. robustness) in a linear relationship** for most dataset shifts. Across CIFAR10 $\ell_\infty$ (Fig. 3), CIFAR10 $\ell_2$ (Fig. 9) and ImageNet $\ell_\infty$ (Fig. 10), the regression of Acc-Acc and Rob-Rob for most shifts achieve very high $R^2$ ($> 0.9$), i.e., their relationship can be well explained by a linear model. This suggests for these shifts ID performance is a good indication of performance under shift, and more importantly, OOD performance can be reliably predicted by ID performance using the fitted linear model.

**Nevertheless, under some shifts, ID and OOD performance are only weakly correlated.** Natural shifts like CIFAR10-R and ImageNet-A and corruption shifts like noise, fog and contrast are observed to have relatively low $R^2$ across varied training set-ups in Figs. 3, 9 and 10. It can be seen from Figs. 15 and 16 that the correlation for these shifts becomes even weaker, and the gap of $R^2$ between them and the others expands, as more inferior (relatively worse accuracy and/or robustness) models are excluded from the regression. This suggests that the models violating the linear trend are mostly high-performance. App. F discusses how inferior models are identified and how they influence the correlation.

The correlation appears to be less related to the degradation, i.e., model performance may fall greatly under some shifts but still correlate well between ID and OOD or vice versa. For example, noise shifts degrade accuracy/robustness mildly in Fig. 2 but have roughly the weakest correlation, whereas CIFAR10-R degrades the most but has a relatively strong correlation.

**AT models exhibit a stronger linear correlation between ID and OOD accuracy.** This is the case under most corruption shifts on CIFAR10 in Figs. 3 and 9. The improvement is dramatic for particular shifts. For example, $R^2$ surges from nearly 0 (no linear correlation) for ST models to around 0.8 (evident linear correlation) for AT models with Gaussian and shot noise data shifts. These results are contrary to the previous finding on ST models (Miller et al., 2021). However, note that we measure linear correlation for the raw data, whereas Miller et al. (2021) applies a nonlinear transformation to the data to promote linearity. Overall, adversarial training boosts linear correlation for corruption shifts, and hence, improves the faithfulness of using ID performance for model selection and OOD performance prediction.

We attribute this to AT improving accuracy on the corrupted data (Kireev et al., 2022). Intuitively, ST models have less correlated corruption accuracy because corruption significantly impairs accuracy and such effect varies a lot among models. Compared to ST, AT effectively mitigates the effect of corruption on accuracy, and hence, reduces the divergence of corruption accuracy so that corruption accuracy is more correlated to ID accuracy.

Last, we observe **no evident correlation when ID and OOD metrics misalign, i.e., Acc-Rob and Rob-Acc for CIFAR10**, but weak correlation for ImageNet $\ell_\infty$ as shown in Fig. 11. This is due

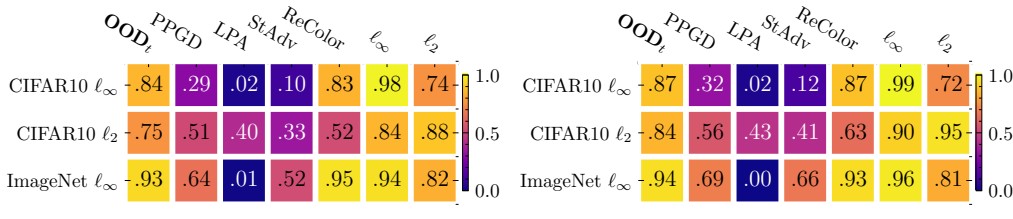

(a) ID seen Rob. vs. ID unforeseen Rob.          (b) $OOD_d$ seen Rob. vs. ID unforeseen Rob.

Figure 4: $R^2$ of regression between seen robustness and unforeseen robustness, i.e., threat shift.

to the inferior models (App. F) on CIFAR-10 which have high accuracy yet poor robustness. They cause OOD robustness to not consistently increase with the ID accuracy. These models are mainly produced by some of our custom training receipts and take a considerable proportion of our CIFAR-10 model zoo, whereas the model zoo of ImageNet is dominated by ones from public sources.

## 4.2 THREAT SHIFT

This section studies the relationship between seen and unforeseen robustness, where seen robustness is computed for both ID and $OOD_d$ data. Unforeseen robustness is computed only using ID data. Linear regression is then conducted between seen robustness ($x$) and unforeseen robustness ($y$). The result of regression for each threat shift is given in App. J. The sensitivity of the regression results to the composition of the model zoo is discussed in App. F.

$\ell_p$ **robustness correlates weakly with non-$\ell_p$ robustness**. The regression between ID seen (i.e., $\ell_p$) robustness and unforeseen robustness (i.e., PPGD, LPA and StAdv) consistently produces $R^2$ below 0.7 across varied training datasets and threat models in Fig. 4a. This means that their relationship cannot be well explained by a linear model. The linear correlation between $\ell_p$ and ReColor is much more evident for $\ell_\infty$ AT models but not $\ell_2$. Nevertheless, $\ell_2$ AT models show a significantly stronger correlation with non-$\ell_p$ robustness than $\ell_\infty$ AT models, except for ReColor. A similar advantage is also observed in ImageNet models when compared to CIFAR10 models.

$\ell_p$ **robustness correlates strongly with $\ell_p$ robustness of different $\epsilon$ and $p$-norm**. $R^2$ of their regression is higher than 0.7 across all assessed set-ups in Fig. 4a suggesting a consistently strong linear correlation. The correlation between different $\epsilon$ of the same $p$-norm is stronger than the correlation between different $p$-norm. Therefore, ID seen $\ell_p$ robustness is useful for selecting a model robust to different $\epsilon$ and $p$-norm attacks, as well as predicting the $\ell_p$ robustness of that model.

**$OOD_d$ seen robustness correlates more strongly than ID seen robustness with unforeseen robustness**. For non-$\ell_p$ threat shifts, the regression w.r.t. $OOD_d$ seen robustness (Fig. 4b) consistently has higher $R^2$ than that of ID seen robustness (Fig. 4a) across all training set-ups. This may imply a deeper underlying relationship between the dataset and the threat shifts.

## 4.3 PREDICTED UPPER LIMIT OF OOD PERFORMANCE

Based on the precise linear trend observed above for existing robust training methods, we can predict the OOD performance of a model trained by such a method from its ID performance using the fitted linear model. Furthermore, we can extrapolate from current trends to predict the maximum OOD robustness that can be expected from a hypothetical future model that achieves perfect robustness on ID data (assuming the linear trend continues).

$$\text{Upper limit of OOD accuracy/robustness (\%)} = \text{slope} * 100 + \text{intercept}. \tag{4}$$

This estimates the best OOD performance one can expect by fully exploiting existing robust training techniques. Note that a wide range of models and techniques (App. C.2) are covered by our correlation analysis so their, as well as their variants', OOD performance should be (approximately) bounded by the predicted upper limit. The accuracy of the prediction depends on the $R^2$ of the corresponding correlation.

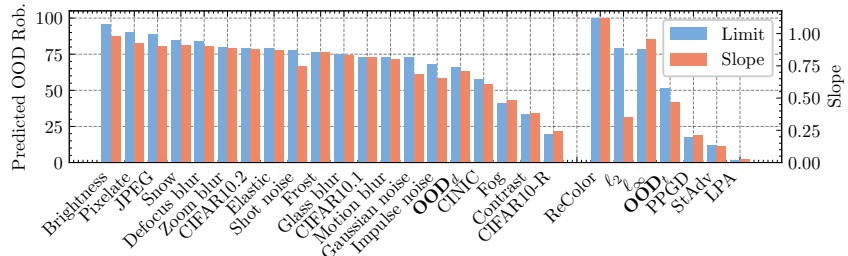

Figure 5: The estimated upper limit of OOD robustness and the conversion rate, a.k.a. slope, to OOD robustness from ID robustness under various distribution shifts for CIFAR10 $\ell_\infty$. The estimated upper limit is capped by 100% as robustness can not surpass 100%.

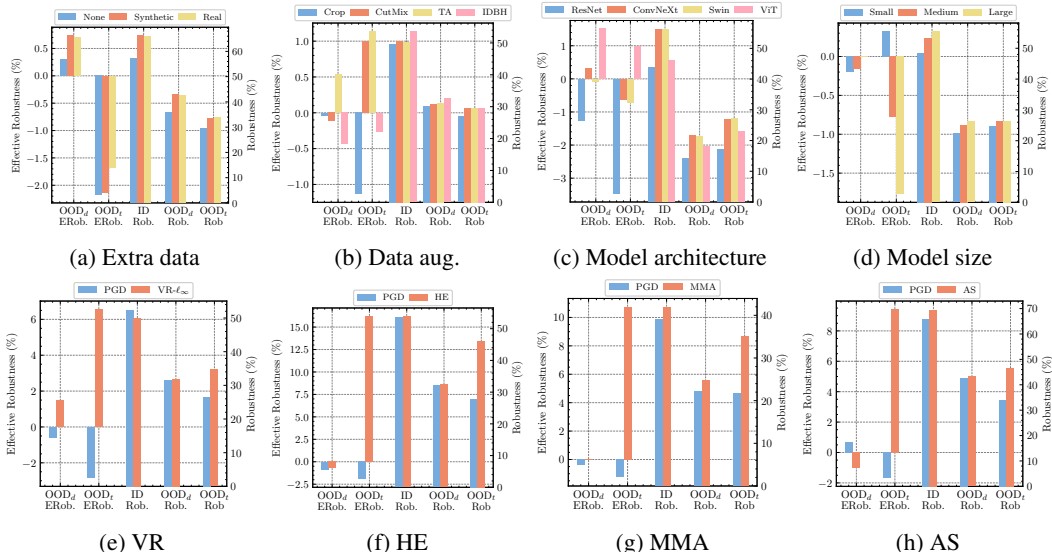

Figure 6: The robustness (Rob.) and adversarial effective robustness (ERob.) of robust techniques.

We find that **continuously improving ID robustness following existing practice may not lead to satisfactory OOD robustness**. The upper limit of OOD robustness under dataset shift, $OOD_d$, is 66%/71%/43% for CIFAR10 $\ell_\infty$ (Fig. 5), CIFAR10 $\ell_2$ (Fig. 13) and ImageNet $\ell_\infty$ (Fig. 14) respectively, and under threat shift $OOD_t$ is 52%/35%/52% correspondingly. Hence, if current trends continue, the resulting models are likely to be very unreliable in real-world applications. The vulnerability of models is most evident for ImageNet $\ell_\infty$ under dataset shift and for CIFAR10 $\ell_2$ under threat shift. The expected upper limit of OOD robustness also varies greatly among individual shifts ranging from nearly 0 to 100%.

One of the accounts for this issue is that the existing methods have poor conversion rate to OOD robustness from ID robustness as shown by the slope of the linear trend in Figs. 5, 13 and 14. Taking an example of fog shift on ImageNet, the slope is roughly 0.1 so improving 10% ID robustness can only lead to 1% improvement on fog robustness. Besides, the upper limit and conversion rate of robustness are observed to be much lower than those of accuracy in Figs. 12 to 14, suggesting the OOD generalization issue is more severe for robustness. Overall, this issue calls for developing novel methods that can improve OOD robustness beyond our prediction.

## 5    IMPROVING OOD GENERALIZATION OF ROBUSTNESS

To inspire the design of methods that have OOD robustness exceeding the above prediction, this section investigates methods that have the potential to be effective for boosting the OOD generalization of robustness. The effectiveness is quantified by two metrics: OOD performance and effective

performance. Effective performance measures the extra resilience of a model under distribution shift when compared to a group of models by adapting the metric of "Effective Robustness" Taori et al. (2020):

$$R'(f) = R_{ood}(f) - \beta(R_{id}(f)) \tag{5}$$

where $\beta(\cdot)$ is a linear mapping from ID to OOD metric fitted on a group of models. We name this metric effective robustness (adversarial effective robustness) when $R_{id}$ and $R_{ood}$ are accuracy (robustness). A positive adversarial effective robustness means that $f$ achieves adversarial robustness above what the linear trend predicts based on its ID performance, i.e., $f$ is advantageous over the fitted models on OOD generalization. Note that higher adversarial effective robustness is not equivalent to higher OOD robustness since the model may have a lower ID robustness.

The specific set-ups and detailed results of the following experiments are described in App. H.

### 5.1 DATA

**Training with extra data** boosts both robustness and adversarial effective robustness for both dataset and threat shifts compared to training schemes without extra data (see Fig. 6a). The improved adversarial effective robustness suggests that this technique induces extra OOD generalizability. There is no clear advantage to training with extra real data (Carmon et al., 2019) rather than synthetic data (Gowal et al., 2021b) except for the adversarial effective robustness under threat shift which is improved more by real data.

**Advanced data augmentation** improves robustness under both types of shifts and adversarial effective robustness under threat shift over the baseline augmentation RandomCrop (see Fig. 6b). Nevertheless, advanced data augmentation methods other than TA (Müller & Hutter, 2021) degrade adversarial effective robustness under dataset shift. TA has inferior ID and OOD robustness than IDBH but consistently achieves the highest adversarial effective robustness.

### 5.2 MODEL

**Advanced model architecture** significantly boosts robustness and adversarial effective robustness under both types of shift over the classical architecture ResNet (He et al., 2016) (Fig. 6c). ConvNeXt (Liu et al., 2022) and Swin (Liu et al., 2021) achieve the highest OOD robustness mainly due to their high ID robustness. Among all tested architectures, ViT (Dosovitskiy et al., 2021) achieves the highest adversarial effective robustness.

**Scaling model up** improves robustness under both types of shift and adversarial effective robustness under dataset shift, but dramatically impairs adversarial effective robustness under threat shift (Fig. 6d). The latter is because increasing model size greatly improves ID robustness but not OOD robustness so that the real OOD robustness is much below the OOD robustness predicted by linear correlation.

### 5.3 TRAINING

**VR** (Dai et al., 2022), the state-of-the-art defense against unforeseen attacks, greatly boosts adversarial effective robustness under threat shifts so that it achieves higher OOD robustness than the baseline in spite of inferior ID robustness. Surprisingly, VR also clearly boosts adversarial effective robustness under dataset shift even though not designed for dealing with these shifts.

Training methods **HS** (Pang et al., 2020), **MMA** (Ding et al., 2020) and **AS** (Bai et al., 2023) achieve an adversarial effective robustness of 16.22%, 10.74% and 9.41%, respectively, under threat shift, which are much higher than corresponding models trained with PGD. Importantly, in contrast to VR, these methods also improve ID robustness resulting in a further boost on OOD robustness. This makes them a potentially promising defense against multi-attack (Dai et al., 2023).

## 6 CONCLUSIONS

This work proposes a new benchmark to assess OOD adversarial robustness, provides many insights into the generalization of existing robust models under distribution shift and identifies several robust

interventions beneficial to OOD generalization. We have analyzed the OOD robustness of hundreds of diverse models to ensure that we obtain generally applicable insights. As we focus on general trends, our analysis does not provide a detailed investigation into individual methods or explain the observed outliers such as the catastrophic robustness degradation. However, OODRobustBench provides a tool for performing such more detailed investigations in the future. It also provides a means of measuring progress towards models that are more robust in real-world conditions and will, hopefully, spur the future development of such models.

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

## A ADDITIONAL COMPARISON WITH RELATED WORKS

### Is the linear trend of robustness really expected given the linear trend of accuracy?

No. There is a well-known trade-off between accuracy and robustness in the ID setting (Tsipras et al., 2019). We further confirm this fact for the models we evaluate in Fig. 11 in the appendix. This means that accuracy and robustness usually go in opposite directions making the linear trend we discover in both particularly interesting. Furthermore, the threat shifts as a scenario of OOD are unique to adversarial evaluation and were thus never explored in the previous studies of accuracy trends.

### How does the linear trends observed by us differ from the previously discovered ones?

Robust models exhibit a stronger linear correlation between ID and OOD accuracy for most corruption shifts (Fig. 3). Particularly, the boost on linearity is dramatic for shifts including Impulse, Shot and Gaussian noises, Glass blur, Pixelate, and JPEG. For instance, R2 surges from 0 (no linear correlation) for non-robust models to 0.84 (evident linear correlation) for robust models with Gaussian noise data shifts. This suggests that, for robust models, predicting OOD accuracy from ID accuracy is more faithful and applicable to more shifts.

The linear trend of robustness is even stronger than that of accuracy for dataset shifts (Fig. 3) but with a lower slope (Sec. 4.3). The latter leads to a predicted upper limit of OOD robustness that is way lower than that of OOD accuracy suggesting that the OOD generalization of robustness is much more challenging.

### How does our analysis differ from the similar analysis in the prior works?

The scale of these previous works is rather small. For instance, RobustBench observes linear correlation only for three shifts on CIFAR-10 based on 39 models with either ResNet or WideResNet

architectures. In such a narrow setting, it is actually neither surprising to see a linear trend nor reliable for predicting OOD performance. By contrast, our conclusion is derived from much more shifts on CIFAR-10 and ImageNet based on 706 models. Importantly, our model zoo covers a diverse set of architectures, robust training methods, data augmentation techniques, and training set-ups. This makes our conclusion more generalizable and the observed (almost perfect) linear trend much more significant.

Similarly, the existing works only test a few models under threat shifts. Those methods are usually just the baseline AT method plus different architectures or the relevant defenses, e.g., jointly trained with multiple threats. It is unclear how the state-of-the-art robust models perform under threat shifts. By conducting a large-scale analysis, we find that those SOTA models generalize poorly to other threats while also discovering several methods that have relatively inferior ID performance but superior OOD robustness under threat shift. Our analysis therefore facilitates future works in this direction by identifying what techniques are ineffective and what are promising.

How does our OOD performance prediction differ from (Baek et al., 2022; Deng & Zheng, 2021; Garg et al., 2021)?

(Baek et al., 2022; Deng & Zheng, 2021; Garg et al., 2021) focus on predicting the OOD accuracy of non-robust models when OOD data is unlabeled, while our work predicts the OOD adversarial robustness of robust models based on labeled OOD data. We quickly summarize these works below:

- (Baek et al., 2022) predicts OOD accuracy based on the linear trend of the prediction agreement of two pairs of models.
- (Garg et al., 2021) learns a threshold for prediction score on labeled ID data and predicts OOD accuracy using the fraction of OOD examples whose prediction score is above the threshold.
- (Deng & Zheng, 2021) predicts OOD accuracy based on a linear trend, learned on the synthetic datasets, between accuracy and distribution shift (measured by Fréchet distance).

The linear trend of adversarial robustness discovered in our work suggests that these methods are potentially applicable to predicting OOD adversarial robustness.

How does you benchmark differ from RobustBench?

Our benchmark focuses on OOD adversarial robustness while RobustBench focuses on ID adversarial robustness. Specifically, our benchmark contrasts RobustBench in the datasets and the attacks. We use CIFAR-10.1, CIFAR-10.2, CINIC, and CIFAR-10-R (ImageNet-V2, ImagetNet-A, ImageNet-R, ObjectNet) to simulate input data distribution shift for the source datasets CIFAR-10 (ImageNet), while RobustBench only uses the latter source datasets. We use PPGD, LPA, ReColor, StAdv, Linf-12/255, L2-0.5 (PPGD, LPA, ReColor, StAdv, Linf-8/255, L2-1) to simulate threat shift for the training threats Linf-8/255 (L2-0.5), while RobustBench only evaluates the same threats as the training ones.

## B  BENCHMARK SET-UP

### B.1  DATASETS

This section introduces the OOD datasets of natural shifts. For ImageNet, we have:

- **ImageNet-V2** is a reproduction of ImageNet using a completely new set of images. It has the same 1000 classes as ImageNet and each class has 10 images so 10K images in total.
- **ImageNet-A** is an adversarially-selected reproduction of ImageNet. The images in this dataset were selected to be those most misclassified by an ensemble of ResNet-50s. It has 200 ImageNet classes and 7.5K images.
- **ImageNet-R** contains various artistic renditions of objects from ImageNet, so there is a domain shift. It has 30K images and 200 ImageNet classes.

- **ObjectNet** is a large real-world dataset for object recognition. It is constructed with controls to randomize background, object rotation and viewpoint. It has 313 classes but only 104 classes compatible with ImageNet classes so we only use this subset. The selected subset includes 17.2K images.

For CIFAR10, we have:

- **CIFAR10.1** is a reproduction of CIFAR10 using a completely new set of images. It has 2K images sampled from the same source as CIFAR10, i.e., 80M TinyImages (Torralba et al., 2008). It has the same number of classes as CIFAR10.

- **CIFAR10.2** is another reproduction of CIFAR10. It has 12K (10k for training and 2k for test) images sampled from the same source as CIFAR10, i.e., 80M TinyImages. It has the same number of classes as CIFAR10. We only use the test set of CIFAR10.2.

- **CINIC** is a downscaled subset of ImageNet with the same image resolution and classes as CIFAR10. Its test set has 90K images in total, of which 20K images are from CIFAR10 and 70K images are from ImageNet. We use only the ImageNet part.

- **CIFAR10-R** is a new dataset created by us. The images in CIFAR10-R and CIFAR10 have different styles so there is a domain shift. We follow the same procedure as CINIC to downscale the images from ImageNet-R to the same resolution as CIFAR10 and select images from the classes of ImageNet corresponding to CIFAR10 classes. We follow the same class mapping between ImageNet and CIFAR10 as CINIC. Note that ImageNet-R does not have images of the ImageNet classes corresponding to CIFAR10 classes of "airplane" and "horse", so there are only 8 classes in CIFAR10-R.

In practice, we evaluate models using a random sample of 5K images from each of the ImageNet variant datasets, and 10K images from each of the CIFAR10 variant datasets, if those datasets contain more images than that number. This is done to accelerate the evaluation and follows the practice used in RobustBench (Croce et al., 2021).

## B.2 ADVERSARIAL EVALUATION

### B.2.1 SEEN ATTACKS

Robustness under dataset shift is evaluated using the same threat model as used in adversarial training, the so-called seen threat model. The perturbation bound $\epsilon$ is 8/255 for CIFAR10 $\ell_\infty$, 0.5 for CIFAR10 $\ell_2$ and 4/255 for ImageNet $\ell_\infty$. We use the MM5 (Gao et al., 2022) attack, instead of other more effective attacks like AutoAttack (Croce & Hein, 2020), for a better balance of efficiency and effectiveness. MM5 is about 32× faster than AutoAttack (Gao et al., 2022). This makes OODRobustBench practical to use and also makes our large-scale analysis on hundreds of models feasible.

To verify the effectiveness of MM5, we compare its result with the result of AutoAttack on the ID dataset across all publicly available models from RobustBench for CIFAR10 $\ell_\infty$, CIFAR10 $\ell_2$ and ImageNet $\ell_\infty$. The gap between the robustness evaluated by them is generally below 0.5%, except for two models: Ding et al. (2020) and Xu et al. (2022). We use MM+ (Gao et al., 2022) attack to evaluate these two models for a more reliable evaluation and the result of MM+ is close to AutoAttack.

### B.2.2 UNFORESEEN ATTACKS

For $\ell_p$ shifts all attacks are performed using MM5. The unforeseen attacks use $\epsilon = 12/255$ and $\epsilon = 0.5$ for $\ell_\infty$ and $\ell_2$ attacks on CIFAR10 $\ell_\infty$, $\epsilon = 8/255$ and $\epsilon = 1$ for $\ell_\infty$ and $\ell_2$ attacks on CIFAR10 $\ell_2$ and on ImageNet $\ell_\infty$.

Ideally, we would like to include more non-$\ell_p$ attacks to understand the generalization of adversarial robustness more thoroughly. However, adding attacks will cause substantial computational costs because of our large model zoo. We choose the current ones because they are sufficiently effective, cover perceptible (ReColor, StAdv) and imperceptible (PPGD, LPA) attacks, and have been widely used (each of them is cited 100+).

# C    MODEL ZOO

## C.1    CRITERIA FOR ROBUST MODELS

We follow the same criteria as the popular benchmarks (RobustBench (Croce et al., 2021), Mul-
tiRobustBench (Dai et al., 2023), etc), which only include robust models that (1) have in general
non-zero gradients w.r.t. the inputs, (2) have a fully deterministic forward pass (i.e. no randomness)
and (3) do not have an optimization loop. These criteria include most AT models, while excluding
most preprocessing methods because they rely on randomness like Guo et al. (2018) or inner opti-
mization loop like Samangouei et al. (2018) which leads to false security, i.e., high robustness to the
non-adaptive attack but vulnerable to the adaptive attack.

Meanwhile, we acknowledge that evaluating dynamic preprocessing-based defenses is still an active
area of research. It is tricky (Croce et al., 2022), and there has not been a consensus on how to
evaluate them. So now, we exclude them for a more reliable evaluation. We will keep maintaining
this benchmark, and we would be happy to include them in the future if the community has reached
a consensus on that (e.g., if these models are merged into RobustBench).

## C.2    MODEL ZOO

Our model zoo consists of 706 models, of which:

- 396 models are trained on CIFAR10 by $\ell_\infty$ 8/255
- 239 models are trained on CIFAR10 by $\ell_2$ 0.5
- 56 models are trained on ImageNet by $\ell_\infty$ 4/255
- 10 models are trained on CIFAR10 for non-$\ell_p$ adversarial robustness
- 5 models are trained on CIFAR10 for common corruption robustness

Among the above models, 66 models of CIFAR10 $\ell_\infty$, 19 models of CIFAR10 $\ell_2$ and 18 models of
ImageNet $\ell_\infty$ are retrieved from RobustBench. 84 models are retrieved from the published works
including Li et al. (2023); Li & Spratling (2023c;b;a); Liu et al. (2023); Singh et al. (2023); Dai et al.
(2022); Hsiung et al. (2023); Mao et al. (2022). The remaining models are trained by ourselves.

We locally train additional models with varying architectures and training parameters to comple-
ment the public models from RobustBench on CIFAR-10. We consider 20 model architectures:
DenseNet-121 (Huang et al., 2017), GoogLeNet (Szegedy et al., 2015), Inception-V3 (Szegedy et al.,
2016), VGG-11/13/16/19 (Simonyan & Zisserman, 2015), ResNet-34/50/101/152 (He et al., 2016),
EfficientNet-B0 (Tan & Le, 2019), MobileNet-V2 (Sandler et al., 2018), DLA (Yu et al., 2018),
ResNeXt-29 (2x64d/4x64d/32x4d/8x64d) (Xie et al., 2017), SeNet-18 (Hu et al., 2018), and Con-
vMixer (Trockman & Kolter, 2023). For each architecture, we vary the training procedure to obtain
15 models across four adversarial training methods: PGD (Madry et al., 2018), TRADES (Zhang
et al., 2019), PGD-SCORE, and TRADES-SCORE (Pang et al., 2022).

We train all models under both $\ell_\infty$ and $\ell_2$ threat models with the following steps:

1. We use PGD adversarial training to train eight models with batch size $\in \{128, 512\}$, a
   learning rate $\in \{0.1, 0.05\}$, and weight decay $\in \{10^{-4}, 10^{-5}\}$. We also save the overall
   best hyperparameter choice. For the $\ell_2$ threat model, we fix the learning rate to 0.1 since
   we observe that with $\ell_\infty$, 0.1 is strictly better than 0.05.
2. Using the best hyperparameter choice, we train one model with PGD-SCORE, three with
   TRADES, and three with TRADES-SCORE. For TRADES and TRADES-SCORE, we
   take their $\beta$ parameter from $0.1, 0.3, 1.0$.

After training, we observe that some locally trained models exhibit inferior accuracy and/or ro-
bustness that is abnormally lower than others. The influence of inferior models on the correlation
analysis is discussed in App. F. Finally, we filter out all models with an overall performance (accu-
racy + robustness) below 110. This threshold is determined to exclude only those evidently inferior
models so that the size of model zoo (557 after filtering) is still large enough to ensure the generality
and comprehensiveness of the conclusions drawn on it.

Table 2: Performance, evaluated with OODRobustBench, of state-of-the-art models trained on CIFAR10 to be robust to $\ell_2$ attacks. Top 3 results under each metric are highlighted by **bold** and/or underscore. The column "OOD" gives the overall OOD robustness which is the mean of the robustness to $OOD_d$ and $OOD_t$.

| Method | Accuracy | | Robustness | | | | Ranking (Rob.) | |
|---|---|---|---|---|---|---|---|---|
| | ID | $OOD_d$ | ID | $OOD_d$ | $OOD_t$ | OOD | ID | OOD |
| Wang et al. (2023) (WRN-70-16) | **95.54** | **80.04** | **84.97** | **60.83** | **36.65** | **48.74** | 1 | 1 |
| Wang et al. (2023) (WRN-28-10) | 95.16 | 79.28 | **83.69** | **59.39** | **35.04** | **47.21** | 2 | 2 |
| Rebuffi et al. (2021) (WRN-70-16-cutmix-extra) | **95.74** | **79.90** | 82.36 | 57.94 | 31.71 | 44.82 | 3 | 4 |
| Gowal et al. (2021a) (extra) | 94.74 | 78.78 | 80.56 | 56.18 | 30.48 | 43.33 | 4 | 6 |
| Rebuffi et al. (2021) (WRN-70-16-cutmix-ddpm) | 92.41 | 75.95 | 80.42 | 56.82 | 34.58 | 45.70 | 5 | 3 |
| Augustin et al. (2020) (WRN-34-10-extra) | 93.97 | 77.40 | 78.81 | 54.71 | 31.62 | 43.16 | 6 | 7 |
| Rebuffi et al. (2021) (WRN-28-10-cutmix-ddpm) | 91.79 | 75.26 | 78.79 | 55.63 | 33.32 | 44.48 | 7 | 5 |
| Sehwag et al. (2022) | 90.93 | 74.00 | 77.29 | 54.33 | 29.44 | 41.88 | 8 | 8 |
| Augustin et al. (2020) (WRN-34-10) | 92.23 | 76.43 | 76.27 | 52.83 | 29.25 | 41.04 | 9 | 11 |
| Rade & Moosavi-Dezfooli (2022) | 90.57 | 73.55 | 76.14 | 53.35 | 29.69 | 41.52 | 10 | 9 |
| Rebuffi et al. (2021) (R18-cutmix-ddpm) | 90.33 | 72.96 | 75.87 | 52.21 | 30.06 | 41.14 | 11 | 10 |
| Gowal et al. (2021a) | 90.89 | 74.71 | 74.51 | 52.20 | 25.76 | 38.98 | 12 | 15 |
| Sehwag et al. (2022) (R18) | 89.76 | 72.31 | 74.42 | 51.76 | 26.68 | 39.22 | 13 | 13 |
| Wu et al. (2020) | 88.51 | 71.23 | 73.66 | 51.53 | 27.50 | 39.52 | 14 | 12 |
| Augustin et al. (2020) | 91.07 | 74.24 | 72.99 | 49.32 | 28.72 | 39.02 | 15 | 14 |
| Engstrom et al. (2019) | 90.83 | 73.85 | 69.25 | 46.65 | 17.71 | 32.18 | 16 | 16 |
| Rice et al. (2020) | 88.67 | 71.27 | 67.69 | 44.76 | 18.58 | 31.67 | 17 | 17 |
| Rony et al. (2019) | 89.04 | 71.77 | 66.46 | 44.54 | 18.31 | 31.42 | 18 | 18 |
| Ding et al. (2020) | 88.00 | 72.32 | 66.09 | 43.79 | 16.52 | 30.15 | 19 | 20 |

Table 3: Performance, evaluated with OODRobustBench, of state-of-the-art models trained on ImageNet to be robust to $\ell_\infty$ attacks. Top 3 results under each metric are highlighted by **bold** and/or underscore. The column "OOD" gives the overall OOD robustness which is the mean of the robustness to $OOD_d$ and $OOD_t$.

| Method | Accuracy | | Robustness | | | | Ranking (Rob.) | |
|---|---|---|---|---|---|---|---|---|
| | ID | $OOD_d$ | ID | $OOD_d$ | $OOD_t$ | OOD | ID | OOD |
| Liu et al. (2023) (Swin-L) | **78.92** | **45.84** | **59.82** | **23.59** | **29.88** | **26.74** | 1 | 1 |
| Liu et al. (2023) (ConvNeXt-L) | **78.02** | **44.74** | **58.76** | **23.35** | **30.10** | **26.72** | 2 | 2 |
| Singh et al. (2023)(ConvNeXt-L-ConvStem) | 77.00 | 44.05 | 57.82 | 23.09 | 27.98 | 25.53 | 3 | 3 |
| Liu et al. (2023) (Swin-B) | 76.16 | 42.58 | 56.26 | 21.45 | 27.02 | 24.24 | 4 | 7 |
| Singh et al. (2023) (ConvNeXt-B-ConvStem) | 75.88 | 42.29 | 56.24 | 21.77 | 27.89 | 24.83 | 5 | 5 |
| Liu et al. (2023) (ConvNeXt-B) | 76.70 | 43.06 | 56.02 | 21.74 | 26.97 | 24.36 | 6 | 6 |
| Singh et al. (2023) (ViT-B-ConvStem) | 76.30 | 44.67 | 54.90 | 21.76 | 28.98 | 25.37 | 7 | 4 |
| Singh et al. (2023) (ConvNeXt-S-ConvStem) | 74.08 | 39.55 | 52.66 | 19.35 | 26.87 | 23.11 | 8 | 9 |
| Singh et al. (2023) (ConvNeXt-B) | 75.08 | 40.68 | 52.44 | 20.09 | 26.06 | 23.07 | 9 | 10 |
| Liu et al. (2023) (Swin-S) | 75.20 | 40.84 | 52.10 | 19.67 | 24.73 | 22.20 | 10 | 12 |
| Liu et al. (2023) (ConvNeXt-S) | 75.64 | 40.91 | 51.66 | 19.40 | 25.00 | 22.20 | 11 | 11 |
| Singh et al. (2023) (ConvNeXt-T-ConvStem) | 72.70 | 38.15 | 49.46 | 17.97 | 25.32 | 21.65 | 12 | 14 |
| Singh et al. (2023) (ViT-S-ConvStem) | 72.58 | 39.24 | 48.46 | 17.83 | 25.43 | 21.63 | 13 | 15 |
| Singh et al. (2023) (ViT-B) | 72.98 | 42.38 | 48.34 | 20.43 | 26.26 | 23.34 | 14 | 8 |
| Debenedetti et al. (2023) (XCiT-L12) | 73.78 | 38.10 | 47.88 | 15.84 | 23.22 | 19.53 | 15 | 18 |
| Singh et al. (2023) (ViT-M) | 71.78 | 39.88 | 47.34 | 18.95 | 25.25 | 22.10 | 16 | 13 |
| Singh et al. (2023) (ConvNeXt-T) | 71.88 | 37.70 | 46.98 | 17.13 | 21.36 | 19.25 | 17 | 19 |
| Mao et al. (2022) (Swin-B) | 74.14 | 38.45 | 46.54 | 15.36 | 22.19 | 18.78 | 18 | 20 |
| Liu et al. (2023)(ViT-B) | 72.84 | 39.88 | 45.90 | 18.01 | 22.95 | 20.48 | 19 | 16 |
| Debenedetti et al. (2023) (XCiT-M12) | 74.04 | 37.00 | 45.76 | 14.73 | 22.82 | 18.77 | 20 | 21 |

# D ADDITIONAL RESULT

## D.1 BENCHMARK

- Tab. 2: benchmark result of state-of-the-art methods for CIFAR10 $\ell_2$.

- Tab. 3: benchmark result of state-of-the-art methods for ImageNet $\ell_\infty$.

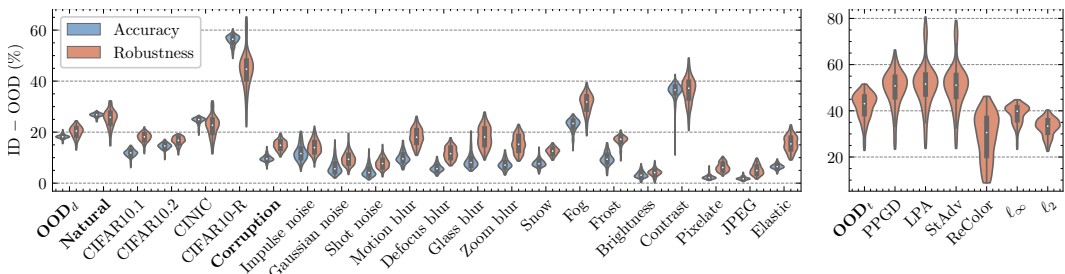

Figure 7: Degradation of accuracy and robustness under various distribution shifts for CIFAR10 $\ell_2$.

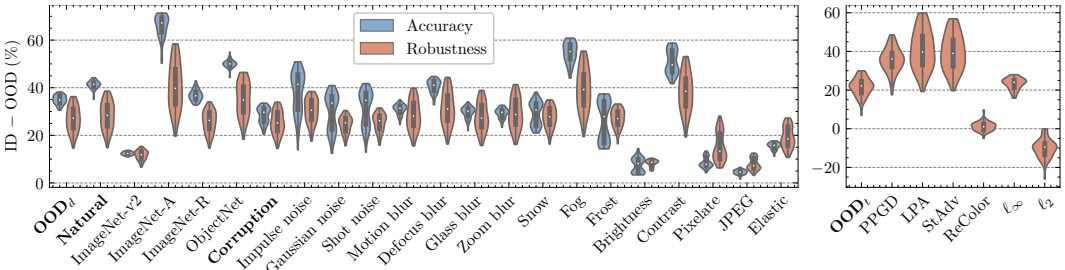

Figure 8: Degradation of accuracy and robustness under various distribution shifts for ImageNet $\ell_\infty$.

## D.2 PERFORMANCE DEGRADATION DISTRIBUTION

- Fig. 7: performance degradation distribution for CIFAR10 $\ell_2$
- Fig. 8: performance degradation distribution for ImageNet $\ell_\infty$.

## D.3 CORRELATION BETWEEN ID AND OOD PERFORMANCE UNDER DATASET SHIFTS

- Fig. 9: $R^2$ of regressions for Acc-Acc and Rob-Rob for CIFAR10 $\ell_2$.
- Fig. 10: $R^2$ of regressions for Acc-Acc and Rob-Rob for ImageNet $\ell_\infty$.
- Fig. 11: $R^2$ of regressions for Acc-Rob and Rob-Acc for CIFAR10 $\ell_\infty$, CIFAR10 $\ell_2$ and ImageNet $\ell_\infty$.

## D.4 PREDICTED UPPER LIMIT OF OOD ACCURACY AND ROBUSTNESS

- Fig. 12: the estimated upper limit of OOD accuracy and the conversion rate for CIFAR10 $\ell_\infty$.
- Fig. 13: the estimated upper limit of OOD performance and the conversion rate for CIFAR10 $\ell_2$.

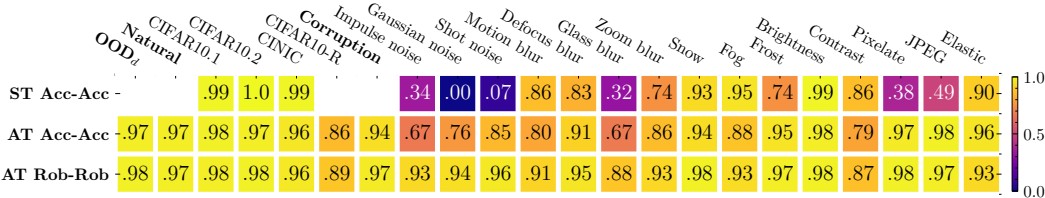

Figure 9: $R^2$ of regression between ID and OOD performance for Standardly-Trained (ST) and Adversarially-Trained (AT) models under various dataset shifts for CIFAR10 $\ell_2$. Higher $R^2$ implies stronger linear correlation. The result of ST models is copied from Miller et al. (2021).

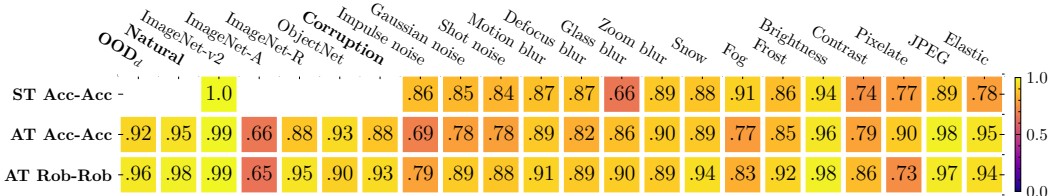

Figure 10: $R^2$ of regression between ID and OOD performance for Standardly-Trained (ST) and Adversarially-Trained (AT) models under various dataset shifts for ImageNet $\ell_\infty$. Higher $R^2$ implies stronger linear correlation. The result of ST models is copied from Miller et al. (2021).

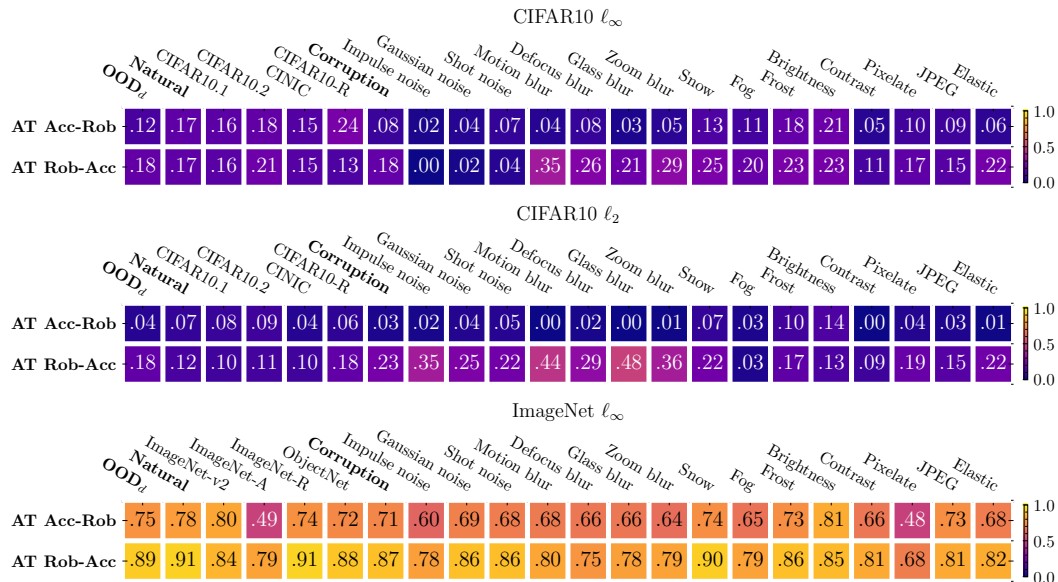

Figure 11: $R^2$ of regression between ID and OOD performance for Adversarially-Trained (AT) models under various dataset shifts. "Acc-Rob" denotes the linear model between ID accuracy (x) and OOD robustness (y) and "Rob-Acc" for ID robustness (x) and OOD accuracy (y).

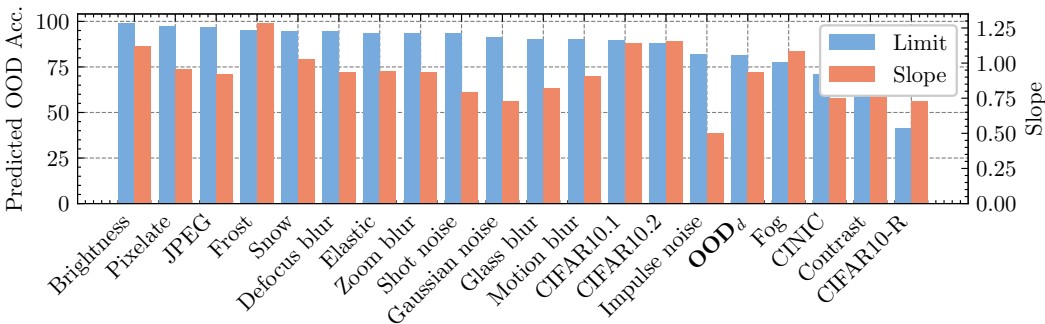

Figure 12: The estimated upper limit of OOD accuracy and the conversion rate, a.k.a. slope, to OOD accuracy from ID accuracy under various distribution shifts for CIFAR10 $\ell_\infty$.

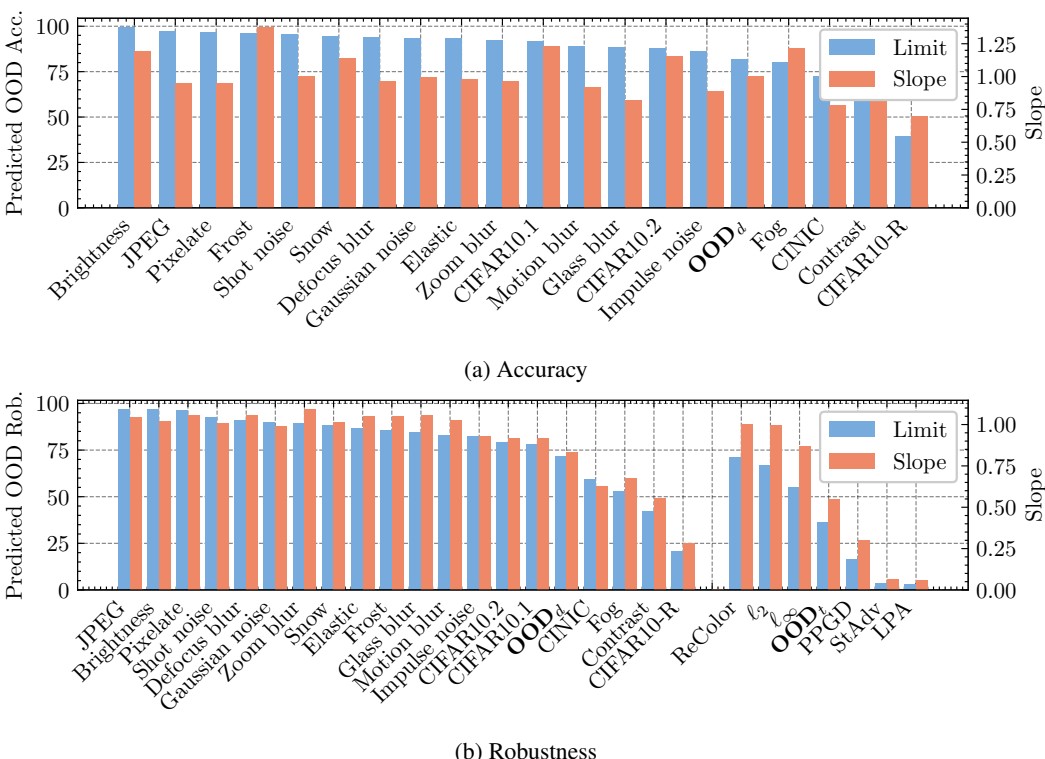

Figure 13: The estimated upper limit of OOD performance and the conversion rate, a.k.a. slope, to OOD performance from ID performance under various distribution shifts for CIFAR10 $\ell_2$.

- Fig. 14: the estimated upper limit of OOD performance and the conversion rate for ImageNet $\ell_\infty$.

# E   CATASTROPHIC DEGRADATION OF ROBUSTNESS

We observe this issue on only one implementation, using WideResNet28-10 with extra synthetic data (model id: *Rade2021Helper_ddpm* on RobustBench), from Rade & Moosavi-Dezfooli (2022) for CIFAR10 $\ell_\infty$. There are three other implementations of this method on RobustBench. None of them, including the one using ResNet18 with extra synthetic data, is observed to suffer from this issue. It seems that catastrophic degradation in this case is specific to the implementation or training dynamics.

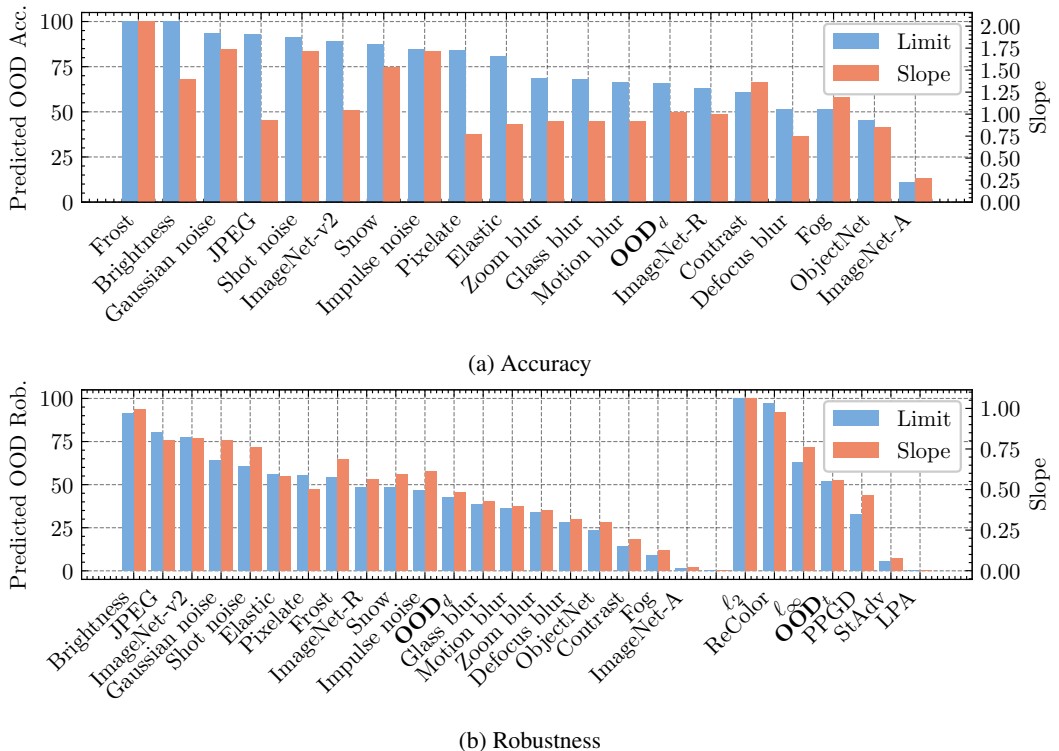

(a) Accuracy

(b) Robustness

Figure 14: The estimated upper limit of OOD performance and the conversion rate, a.k.a. slope, to OOD performance from ID performance under various distribution shifts for ImageNet $\ell_\infty$.

On the other hand, catastrophic degradation consistently happens on the models trained with AutoAugment or IDBH but not other tested data augmentations. It suggests the possibility that a certain image transformation operation exclusively used by AutoAugment and IDBH cause this issue. Besides, catastrophic degradation also consistently happens on the models trained using the receipt of Debenedetti et al. (2023) under Gaussian and shot noise shifts. However, it employs a wide range of training techniques, so further experiments are required to identify the specific cause.

## F HOW INFERIOR MODELS AFFECT THE CORRELATION ANALYSIS

This section studies the influence of the construction of model zoo on the result of correlation. We use the overall performance (accuracy + robustness) to filter out inferior models. As we increasing the threshold of overall performance for filtering, the average overall performance of the model zoo increases, the number of included models decreases and the weight of the models from other published sources on the regression grows up. Our locally trained models are normally inferior to the public models regarding the performance since the latter employs better optimized and more effective training methods and settings. The training methods and settings of public models are also much more diverse.

The correlation for particular shifts varies considerably as more inferior models removed. $R^2$ declines considerably under CIFAR10-R, noise, fog, glass blur, frost and contrast for both Acc-Acc and Rob-Rob on CIFAR10 $\ell_\infty$ (Fig. 15) and $\ell_2$ (Fig. 16). A similar trend is also observed for threat shifts, ReColor and different $p$-norm for CIFAR10 $\ell_\infty$ as shown in Fig. 17. It suggests that the weak correlation under these shifts mainly results from those high-performance public models, and is likely related to the fact that these models include much diverse training methods and settings. For example, all observed catastrophic degradation under the noise shifts occur in the public models. Note that the locally trained models have a large diversity in model architectures particularly within the family of CNNs, but it seems that this architectural diversity does not effect the correlation as much as other factors.

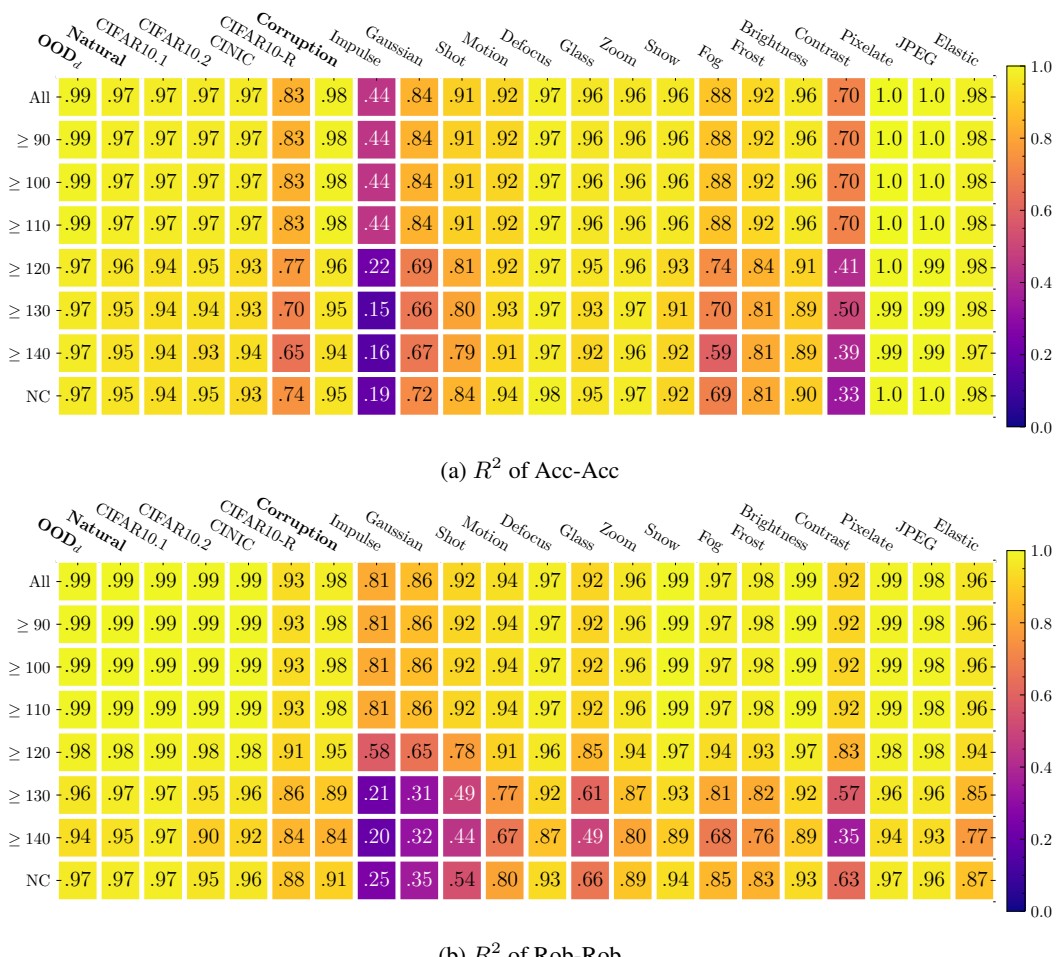

Figure 15: How $R^2$ under various dataset shifts changes as the models with lower overall performance are removed from regression for CIFAR10 $\ell_\infty$. Each row, with the filtering threshold labeled at the lead, corresponds to a new filtered model zoo and the regression conducted it. "NC" refers to No Custom models, so all models are retrieved from either RobustBench or other published works.

In contrast, correlation is improved for most threat shifts for CIFAR10 $\ell_2$ as shown in Fig. 17. As shown in Fig. 26, the locally trained (inferior) models and the public (high-performance) models have divergent linear trends (most evident in the plot of PPGD). That's why removing models from either group will enhance the correlation. Note that such divergence is not evident in the figures of CIFAR10 $\ell_\infty$ (Fig. 25) and ImageNet $\ell_\infty$ (Fig. 27).

# G   PREDICTING PERFORMANCE ON UNLABELED OOD DATA

It is challenging to apply (Baek et al., 2022; Deng & Zheng, 2021; Garg et al., 2021) to predict OOD adversarial robustness with only unlabeled OOD data because the generation of strong OOD adversarial examples typically requires labels. Supposing we have already generated these adversarial examples, we run the experiments for CIFAR-10 models with CIFAR-10.1 and Impulse noise shifts and find that a linear trend is also observed in the agreement between the predictions of any two pairs of robust models: R2 is 0.95 for Impulse noise shift and 0.99 for CIFAR-10.1 shift. This suggests that the method (Baek et al., 2022) is also effective in predicting OOD adversarial robustness.

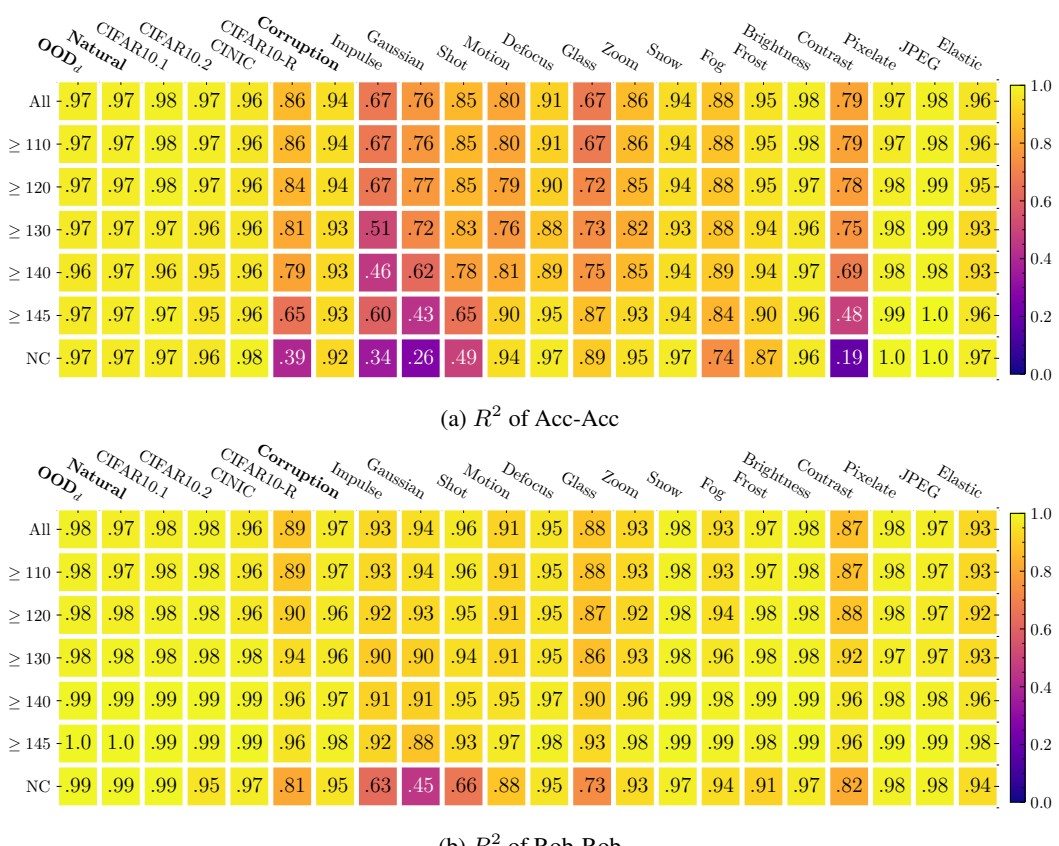

Figure 16: How $R^2$ under various dataset shifts changes as the models with lower overall performance are removed from regression for CIFAR10 $\ell_2$. Each row, with the filtering threshold labeled at the lead, corresponds to a new filtered model zoo and the regression conducted it. "NC" refers to No Custom models, so all models are retrieved from either RobustBench or other published works.

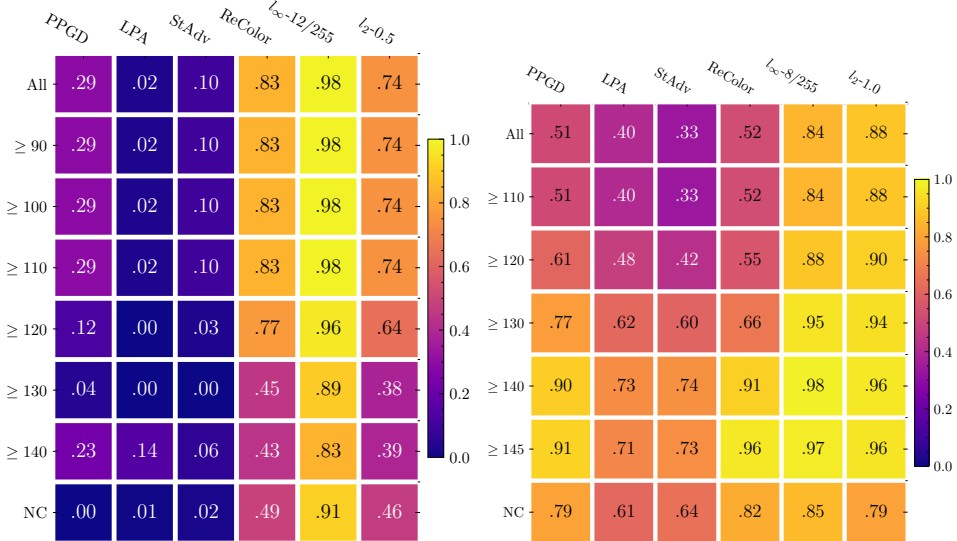

(a) $R^2$ of ID seen vs. ID unforeseen robustness for CIFAR10 $\ell_\infty$

(b) $R^2$ ID seen vs. ID unforeseen robustness for CIFAR10 $\ell_2$

Figure 17: How $R^2$ under various threat shifts changes as the models with lower overall performance are removed from regression. Each row, with the filtering threshold labeled at the lead, corresponds to a new filtered model zoo and the regression conducted it. "NC" refers to No Custom models, so all models are retrieved from either RobustBench or other published works.

Table 4: The effect of training with extra data on the OOD generalization of accuracy and robustness.

| Dataset | Threat Model | Training | Model Architecture | Extra Data | ID Acc. | ID Rob. | OOD$_d$ Acc. | OOD$_d$ Rob. | OOD$_d$ EAcc. | OOD$_d$ ERob. | OOD$_t$ Rob. | OOD$_t$ ERob. |
|---|---|---|---|---|---|---|---|---|---|---|---|---|
| CIFAR10 | Linf | Gowal et al. (2021a) | WideResNet70-16 | - | 85.29 | 57.24 | 66.98 | 35.90 | -0.56 | 0.30 | 29.39 | -2.18 |
| | | | | Synthetic | 88.74 | **66.24** | 70.68 | **42.76** | -0.08 | **0.74** | 33.65 | -2.13 |
| | | | | Real | **91.10** | 66.03 | **73.24** | 42.58 | **0.26** | 0.71 | **34.00** | **-1.67** |

## H  EFFECTIVE ROBUST INTERVENTION

All models used in this analysis are retrieved from RobustBench or other published works to ensure they are well-trained by the techniques to be examined. For each robust intervention, some general training setting, the reference to the source of models and the detailed performance are summarized in the following tables:

- Tab. 4: training with extra data.
- Tab. 5: training with advanced data augmentation.
- Tab. 6: training with advanced model architectures.
- Tab. 7: scaling models up.
- Tab. 8: training techniques of VR, HE, MMA and AS.

The specific experiment setting for each model can be found in its original paper.

### H.1  THE IMPACT OF OOD GENERALIZATION METHODS

Our conclusion on linear trend and upper limit mainly applies to existing AT methods and enhancement techniques such as extra data, model scaling, etc. Since OOD generalization methods, such as, IRM (Arjovsky et al., 2019) and REx (Krueger et al., 2021), are rarely used in the existing AT methods, their robustness to attacks is unclear and so not reflected in our conclusions. In fact, one

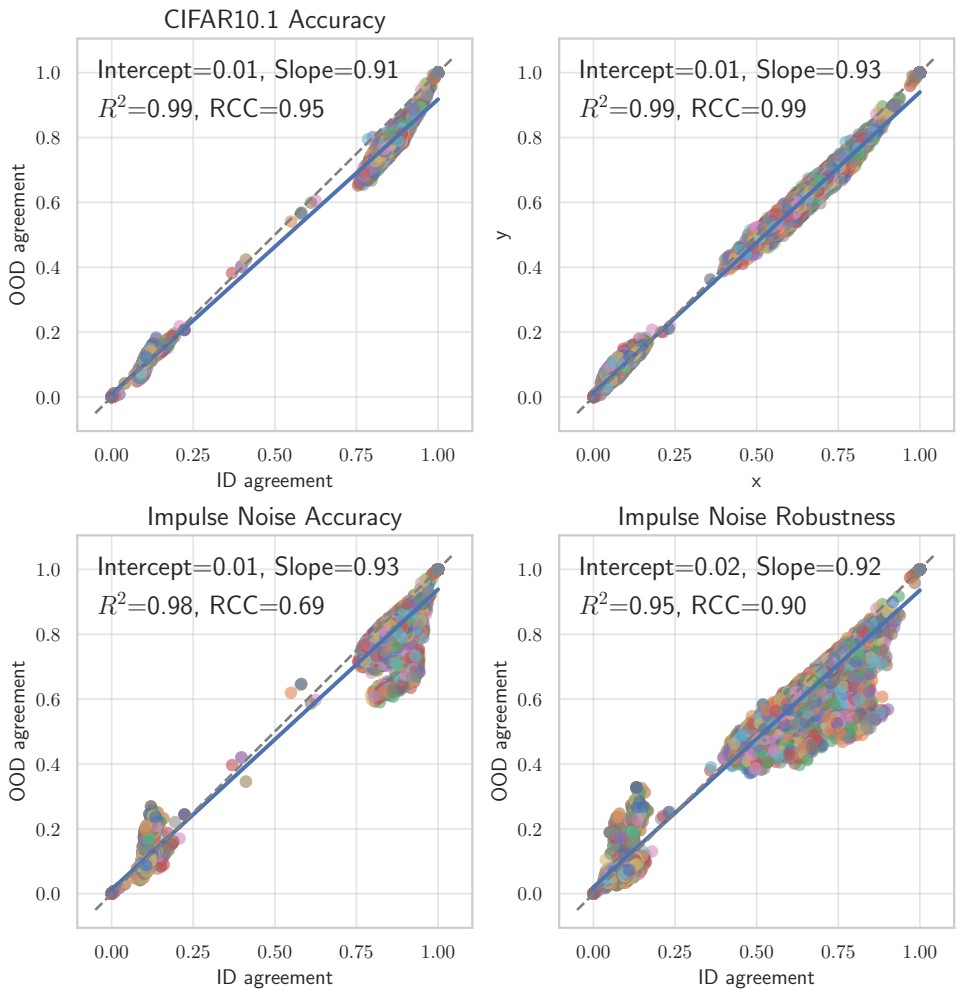

Figure 18: Correlation between seen and unforeseen robustness on ID data for CIFAR10 $\ell_2$ AT models

Table 5: The effect of data augmentation on the OOD generalization of accuracy and robustness. The results reported in Fig. 6b are the mean of the results on ViT and WideResNets.

| Dataset | Threat Model | Training | Model Architecture | Data Augmentation | ID Acc. | ID Rob. | $OOD_d$ Acc. | $OOD_d$ Rob. | $OOD_d$ EAcc. | $OOD_d$ ERob. | $OOD_t$ Rob. | $OOD_t$ ERob. |
|---|---|---|---|---|---|---|---|---|---|---|---|---|
| | | | | RandomCrop | 83.23 | 47.02 | 66.48 | 28.85 | 0.86 | 0.54 | 27.36 | 0.57 |
| | | | | Cutout | **84.22** | **49.57** | **67.23** | **30.68** | 0.69 | 0.56 | 29.74 | 1.75 |
| | | | ViT-B | CutMix | 80.92 | 47.45 | 63.93 | 29.89 | 0.48 | **1.27** | 30.48 | 3.49 |
| | | | | TrivialAugment | 80.33 | 46.61 | 64.59 | 29.56 | **1.69** | **1.54** | 30.40 | **3.80** |
| | | | | AutoAugment | 82.75 | 48.11 | 65.89 | 29.78 | 0.73 | 0.69 | **30.90** | 3.60 |
| | | Li & | | IDBH | **86.92** | **51.55** | **70.51** | **32.08** | 1.45 | 0.54 | **30.59** | 1.68 |
| CIFAR10 | Linf | Spratling (2023c) | | RandomCrop | 86.52 | 52.42 | 68.11 | 31.55 | -0.58 | **-0.61** | 26.47 | -2.84 |
| | | | | Cutout | 86.77 | 53.31 | 68.40 | 31.03 | -0.53 | -1.76 | 27.00 | -2.74 |
| | | | WideResNet34-10 | CutMix | 87.41 | 53.89 | 68.97 | 31.71 | -0.55 | -1.50 | 28.50 | **-1.50** |
| | | | | TrivialAugment | 86.98 | 54.18 | 69.85 | **32.94** | 0.73 | **-0.47** | 28.62 | -1.52 |
| | | | | AutoAugment | **87.93** | **55.10** | **70.05** | 32.17 | 0.04 | -1.90 | **29.06** | -1.51 |
| | | | | IDBH | **88.62** | **55.56** | **70.96** | **32.99** | 0.30 | -1.41 | 28.58 | -2.21 |

Table 6: The effect of model architecture on the OOD generalization of accuracy and robustness.

| Dataset | Threat Model | Training | Model Architecture | Model Size (M) | ID Acc. | ID Rob. | OOD$_d$ Acc. | OOD$_d$ Rob. | OOD$_d$ EAcc. | OOD$_d$ ERob. | OOD$_t$ Rob. | OOD$_t$ ERob. |
|---|---|---|---|---|---|---|---|---|---|---|---|---|
| ImageNet | $\ell_\infty$ | Liu et al. (2023) | ResNet152 | 60.19 | 70.92 | 43.62 | 34.43 | 14.13 | -1.71 | -1.26 | 17.23 | -3.47 |
| | | | ConvNeXt-B | **88.59** | **76.70** | 56.02 | **43.06** | **21.74** | 1.03 | 0.33 | 26.97 | -0.63 |
| | | | ViT-B | 86.57 | 72.84 | 45.90 | 39.88 | 18.01 | **1.78** | **1.51** | 22.95 | **0.98** |
| | | | Swin-B | 87.77 | 76.16 | **56.26** | 42.58 | 21.45 | 1.10 | -0.07 | **27.02** | -0.72 |

Table 7: The effect of model size on the OOD generalization of accuracy and robustness. The results reported in Fig. 6d are averaged over three architectures at the corresponding relatively model size. For example, the result of "small" is averaged over WideResNet28-10, ResNet50 and ConvNeXt-S-ConvStem.

| Dataset | Threat Model | Training | Model Architecture | Model Size | ID Acc. | ID Rob. | OOD$_d$ Acc. | OOD$_d$ Rob. | OOD$_d$ EAcc. | OOD$_d$ ERob. | OOD$_t$ Rob. | OOD$_t$ ERob. |
|---|---|---|---|---|---|---|---|---|---|---|---|---|
| CIFAR10 | $\ell_\infty$ | Rebuffi et al. (2021) | WideResNet28-10 | 36.48 | 87.33 | 60.88 | 69.35 | 38.54 | -0.10 | 0.35 | 33.63 | **0.36** |
| | | | WideResNet70-16 | 266.80 | 88.54 | 64.33 | 70.62 | 41.01 | 0.04 | 0.35 | **34.12** | -0.76 |
| | | | WideResNet106-16 | **415.48** | **88.50** | **64.82** | **70.65** | **41.43** | **0.11** | **0.42** | 33.90 | -1.22 |
| ImageNet | $\ell_\infty$ | Liu et al. (2023) | ResNet50 | 25.56 | 65.02 | 32.02 | 28.43 | 9.23 | **-1.68** | **-0.53** | 13.71 | **-0.52** |
| | | | ResNet101 | 44.55 | 68.34 | 39.76 | 31.74 | 12.44 | -1.76 | -1.08 | 16.82 | -1.72 |
| | | | ResNet152 | **60.19** | **70.92** | **43.62** | **34.43** | **14.13** | -1.71 | -1.26 | **17.23** | -3.47 |
| ImageNet | $\ell_\infty$ | Singh et al. (2023) | ConvNeXt-S-ConvStem | 50.26 | 74.08 | 52.66 | 39.55 | 19.35 | 0.19 | -0.42 | 26.87 | 1.14 |
| | | | ConvNeXt-B-ConvStem | 88.75 | 75.88 | 56.24 | 42.29 | 21.77 | 1.10 | 0.26 | 27.89 | 0.16 |
| | | | ConvNeXt-L-ConvStem | **198.13** | **77.00** | **57.82** | **44.05** | **23.09** | **1.71** | **0.80** | **27.98** | -0.63 |

Table 8: The effect of different adversarial training methods on the OOD generalization of accuracy and robustness.

| Dataset | Threat | Training | ID Acc. | ID Rob. | OOD$_d$ Acc. | OOD$_d$ Rob. | OOD$_d$ EAcc. | OOD$_d$ ERob. | OOD$_t$ Rob. | OOD$_t$ ERob. |
|---|---|---|---|---|---|---|---|---|---|---|
| CIFAR10 | $\ell_\infty$ | PGD (Li & Spratling, 2023c) | **86.52** | **52.42** | **68.11** | 31.55 | -0.58 | -0.61 | 26.47 | -2.84 |
| | | VR-$\ell_\infty$ (Dai et al., 2022) | 72.72 | 49.92 | 56.12 | **31.84** | 0.34 | **1.47** | **34.70** | **6.55** |
| | | PGD (Rice et al., 2020) | **85.34** | 53.52 | 66.46 | 32.07 | -1.12 | -0.88 | 27.89 | -1.94 |
| | | HE (Pang et al., 2020) | 85.14 | **53.84** | **66.96** | **32.45** | **-0.43** | **-0.72** | **46.20** | **16.22** |
| | | PGD (locally-trained) | 80.44 | 38.98 | 62.40 | 22.18 | -0.60 | -0.39 | 21.77 | -1.27 |
| | | MMA (Ding et al., 2020) | **84.37** | **41.86** | **68.22** | **24.65** | **1.54** | **0.02** | **35.12** | **10.74** |
| | | PGD Gowal et al. (2021a) | 91.10 | 66.03 | 73.24 | 42.58 | 0.26 | **0.71** | 34.00 | -1.67 |
| | | AS Bai et al. (2023) | **95.23** | **69.50** | **79.09** | **43.32** | **2.25** | -1.03 | **46.71** | **9.41** |

Table 9: The effect of self-supervised pre-training, ACL, on the OOD generalization of accuracy and robustness.

| Dataset | Threat Model | Training | Model Architecture | ID Acc. | ID Rob. | OOD$_d$ Acc. | OOD$_d$ Rob. | OOD$_d$ EAcc. | OOD$_d$ ERob. | OOD$_t$ Rob. | OOD$_t$ ERob. |
|---|---|---|---|---|---|---|---|---|---|---|---|
| CIFAR10 | $\ell_\infty$ | ACL | ResNet18 | 82.31 | 49.38 | 64.19 | 30.11 | -0.56 | 0.12 | 28.53 | 0.64 |

of the implications of this work is to hopefully spur the future development of new approaches that can cope with distribution shifts better and achieve OOD robustness beyond our prediction.

The OOD generalization methods alone are unlikely to increase the limit because they offer little or no adversarial robustness (see the OOD robustness of leading corruption defenses from Robust-Bench in Tab. 1). Whether an OOD generalization method (when combined with adversarial training) can improve the OOD robustness likely differs from one method to another. On the one hand, it is observed in (Ibrahim et al., 2023) that variance-based risk extrapolation (REx), when combined with adversarial training, improves OOD adversarial robustness under threat shifts.

On the other hand, regarding unsupervised representation learning, the methods listed in (Shen et al., 2021) have never been applied before to adversarial training so it is unclear whether they will help or not. There is a line of work on self-supervised adversarial training which is close to (Shen et al., 2021). We have evaluated one of these methods named Adversarial Contrastive Learning (ACL) (Jiang et al., 2020) which combines self-supervised contrastive learning with adversarial training. The effective robustness under dataset shift and threat shift is 0.12% and 0.64% (Tab. 9), suggesting only marginal benefit in improving the limit of OOD robustness.

# I  PLOTS OF CORRELATION PER DATASET SHIFT

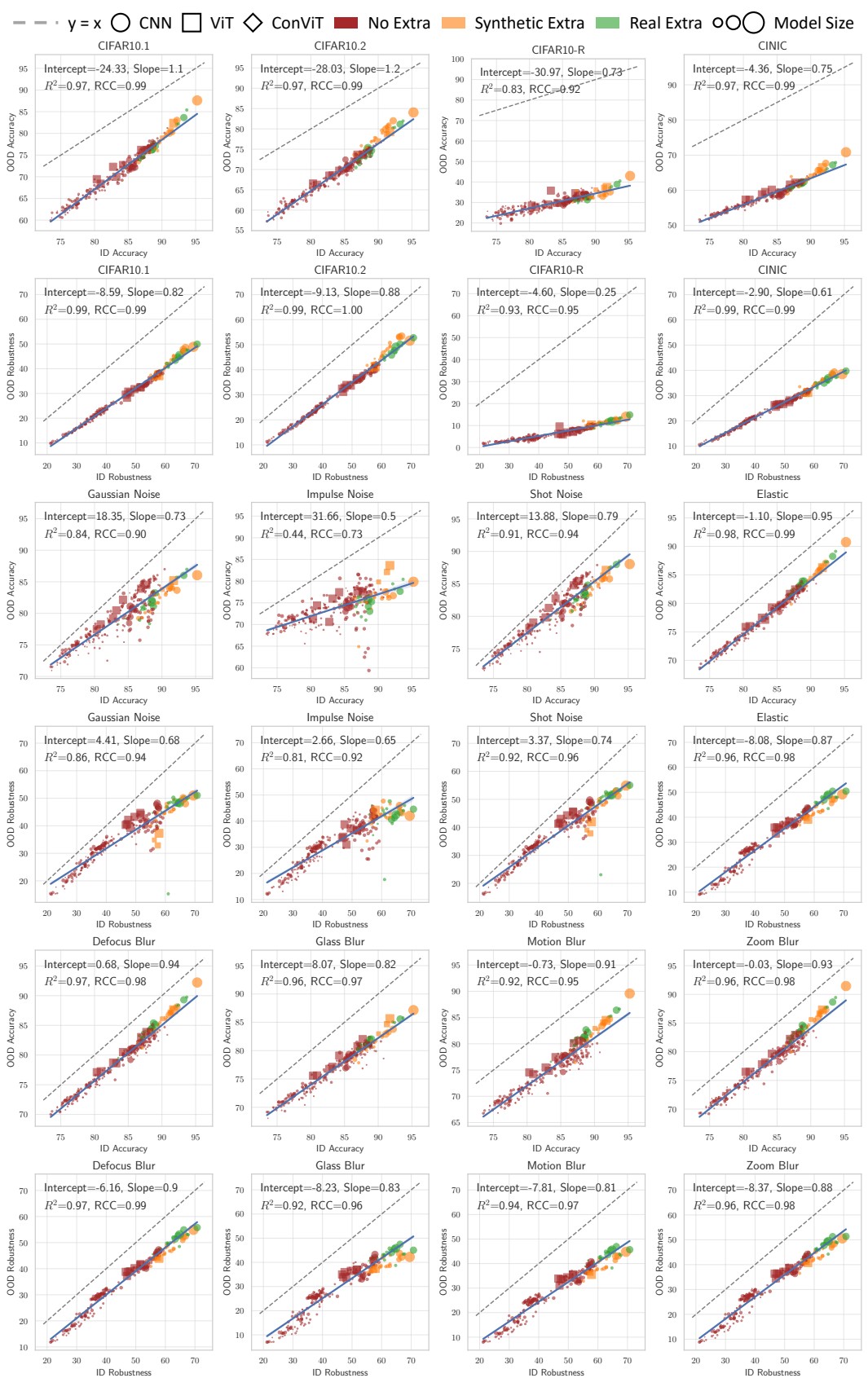

Figure 19: Correlation between ID accuracy and OOD accuracy (odd rows); ID robustness and OOD robustness (even rows) for CIFAR10 $\ell_\infty$ AT models

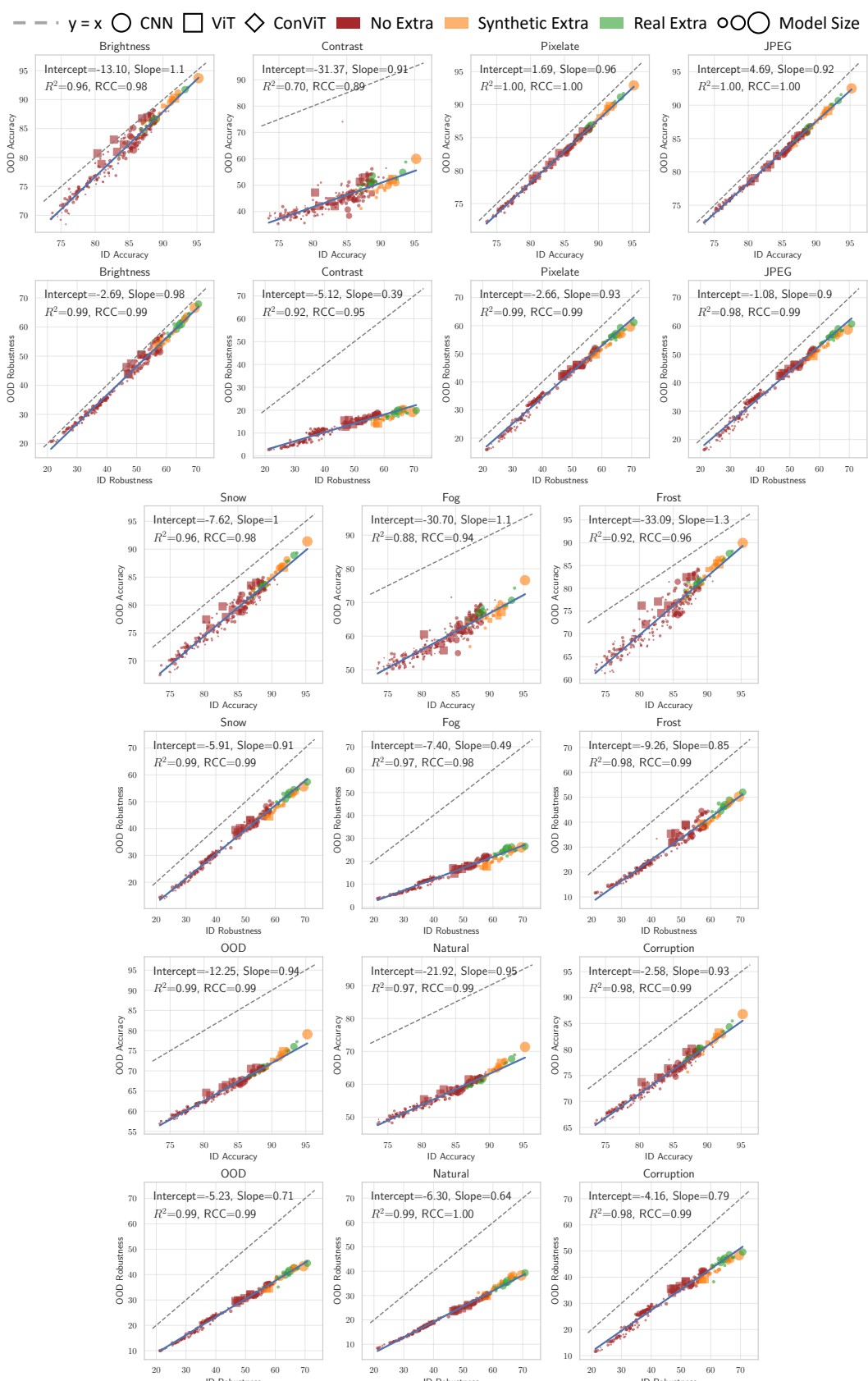

Figure 20: Correlation between ID accuracy and OOD accuracy (odd rows); ID robustness and OOD robustness (even rows) for CIFAR10 $\ell_\infty$ AT models

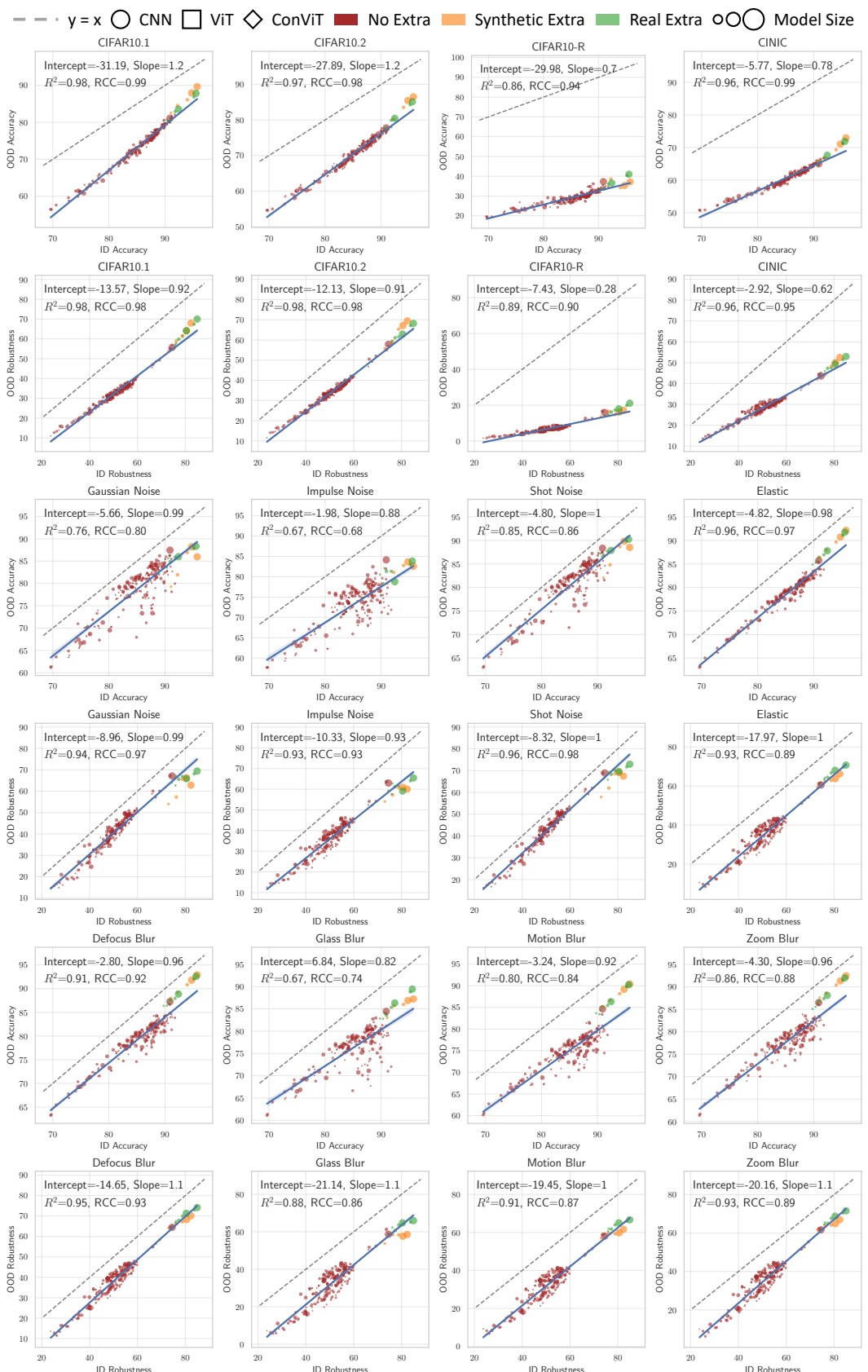

Figure 21: Correlation between ID accuracy and OOD accuracy (odd rows); ID robustness and OOD robustness (even rows) for CIFAR10 $\ell_2$ AT models

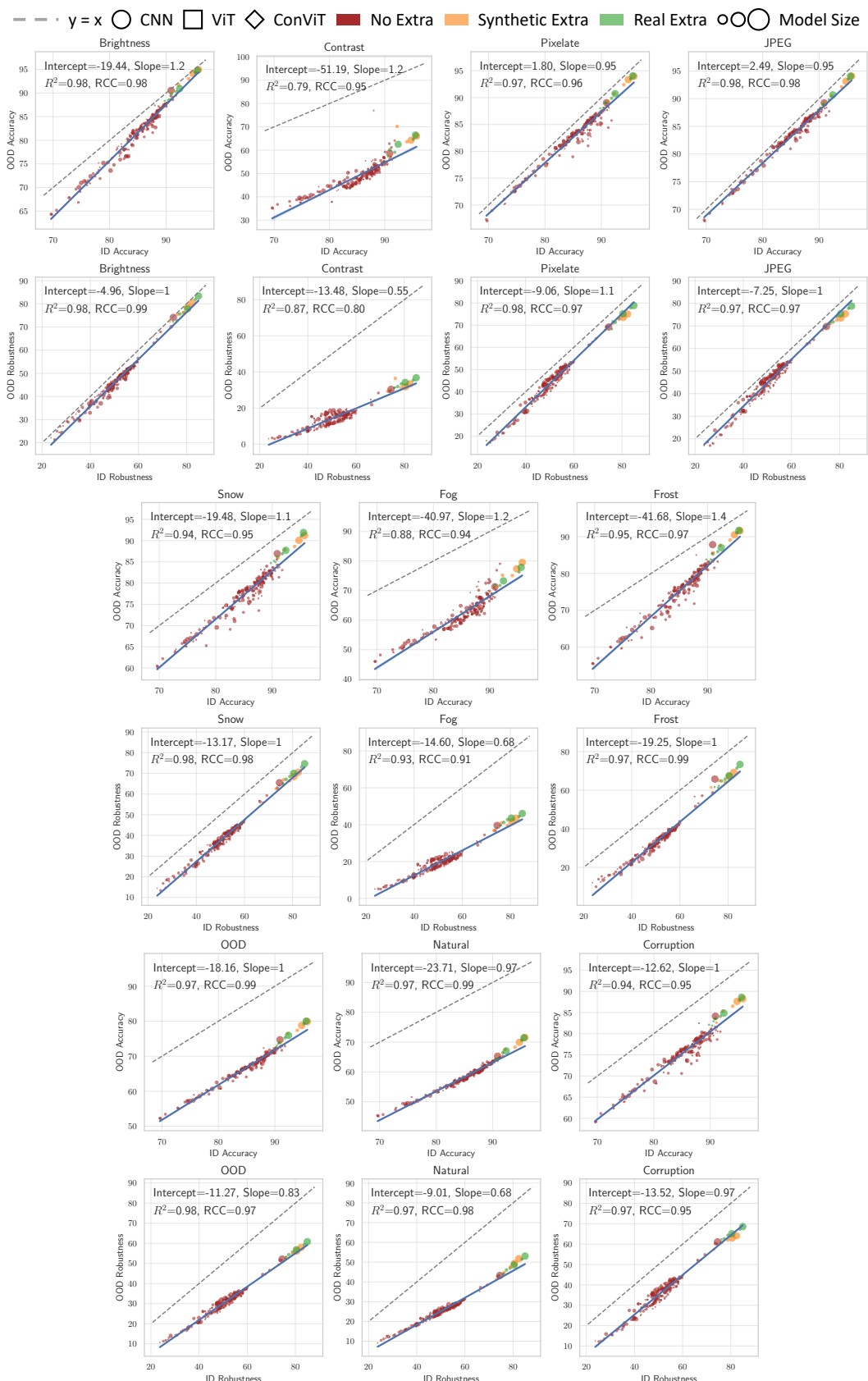

Figure 22: Correlation between ID accuracy and OOD accuracy (odd rows); ID robustness and OOD robustness (even rows) for CIFAR10 $\ell_2$ AT models

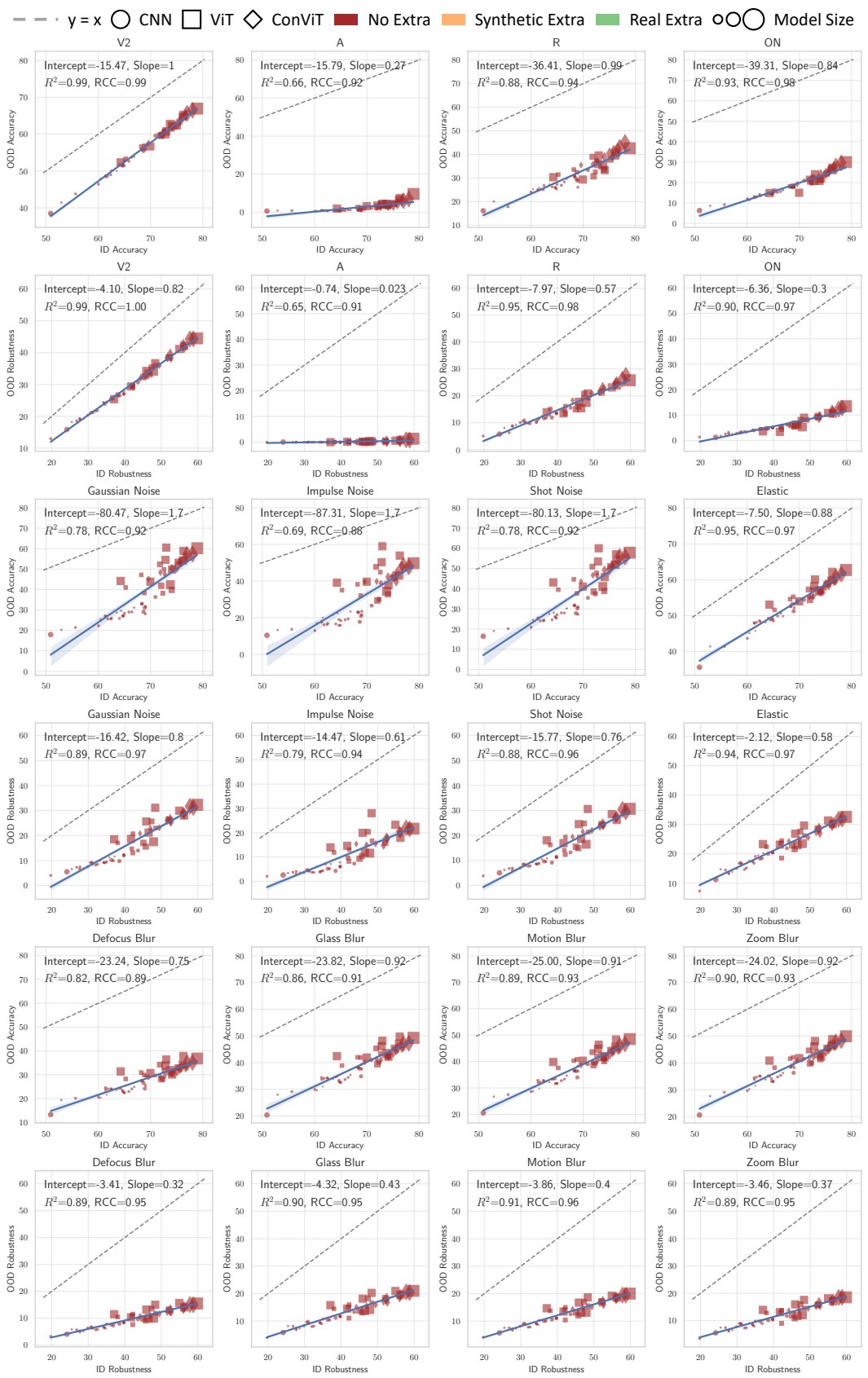

Figure 23: Correlation between ID accuracy and OOD accuracy (odd rows); ID robustness and OOD robustness (even rows) for ImageNet $\ell_\infty$ AT models

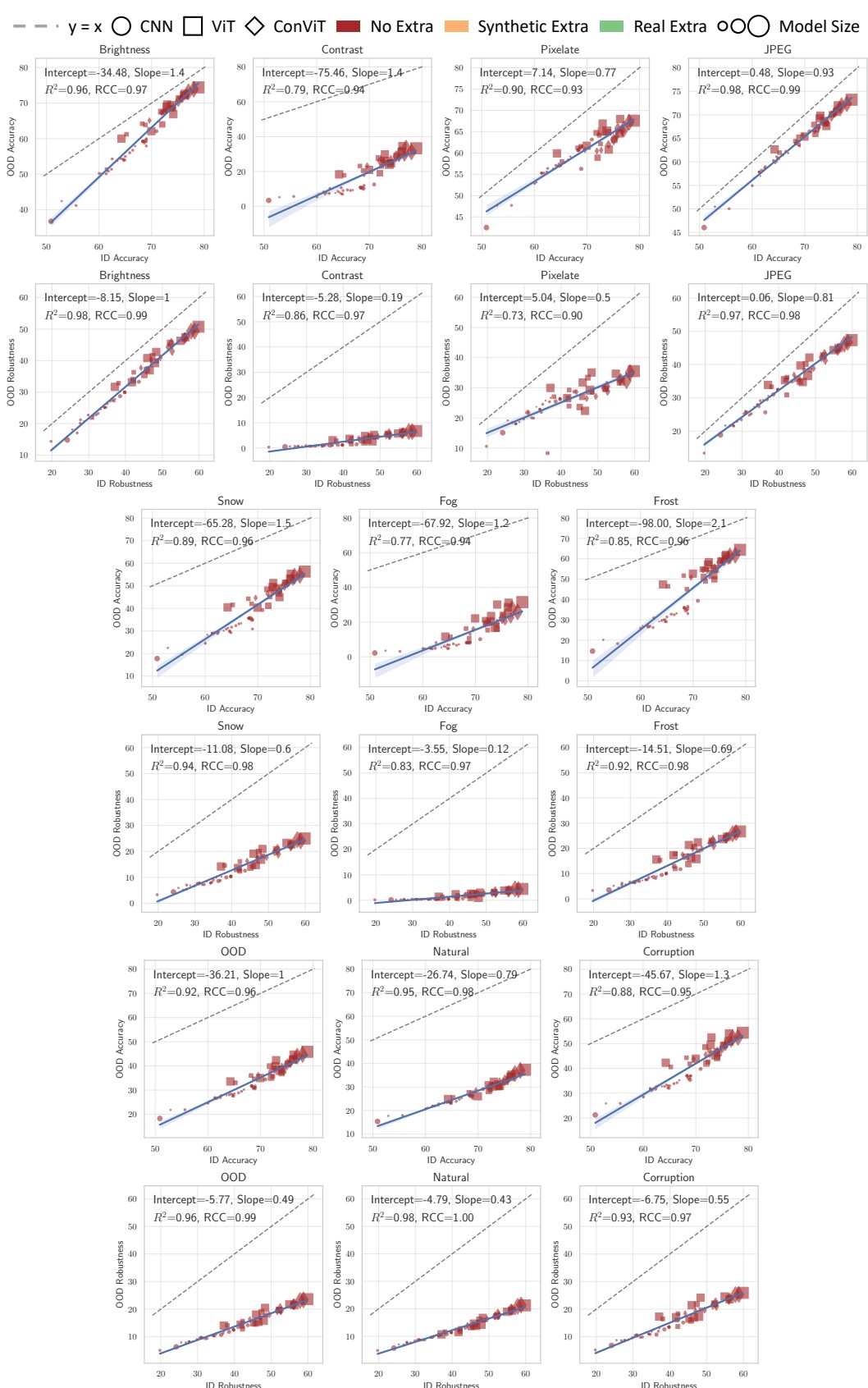

Figure 24: Correlation between ID accuracy and OOD accuracy (odd rows); ID robustness and OOD robustness (even rows) for ImageNet $\ell_\infty$ AT models

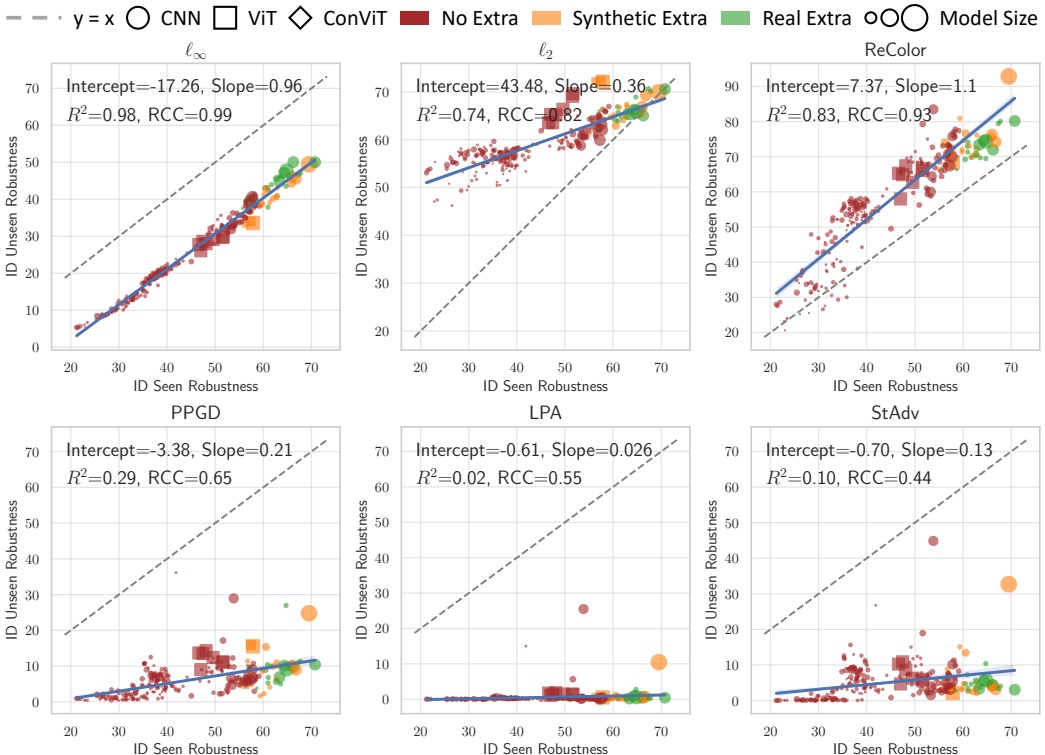

Figure 25: Correlation between seen and unforeseen robustness on ID data for CIFAR10 $\ell_\infty$ AT models

## J    PLOTS OF CORRELATION PER THREAT SHIFT

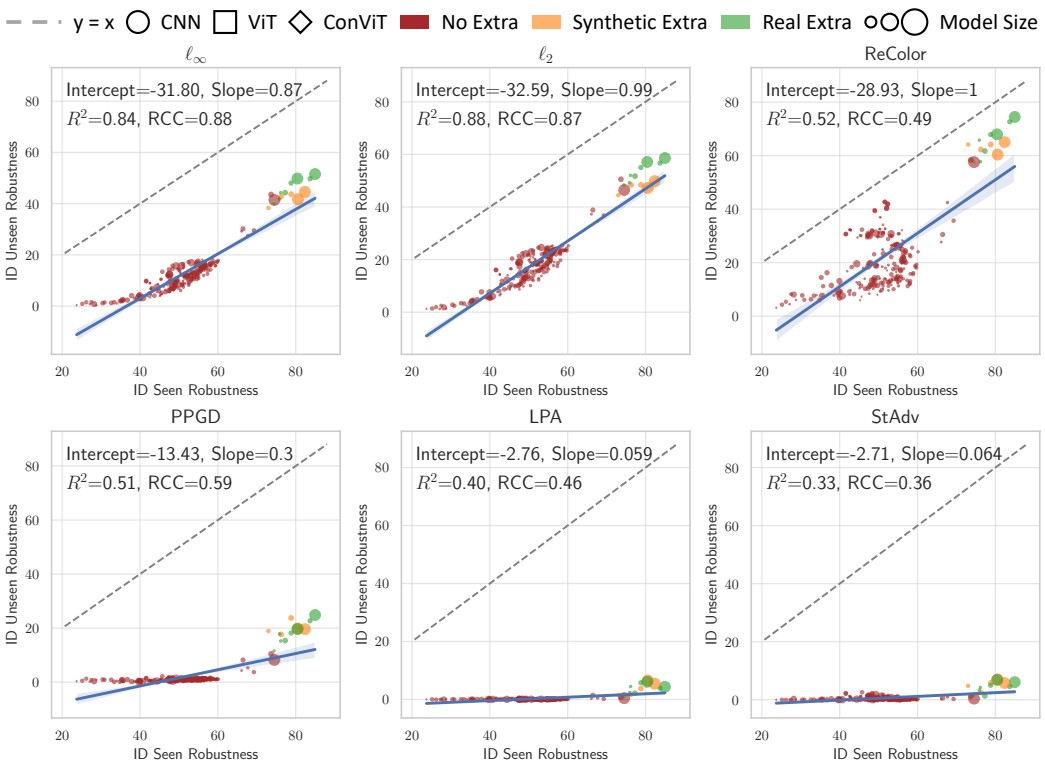

Figure 26: Correlation between seen and unforeseen robustness on ID data for CIFAR10 $\ell_2$ AT models

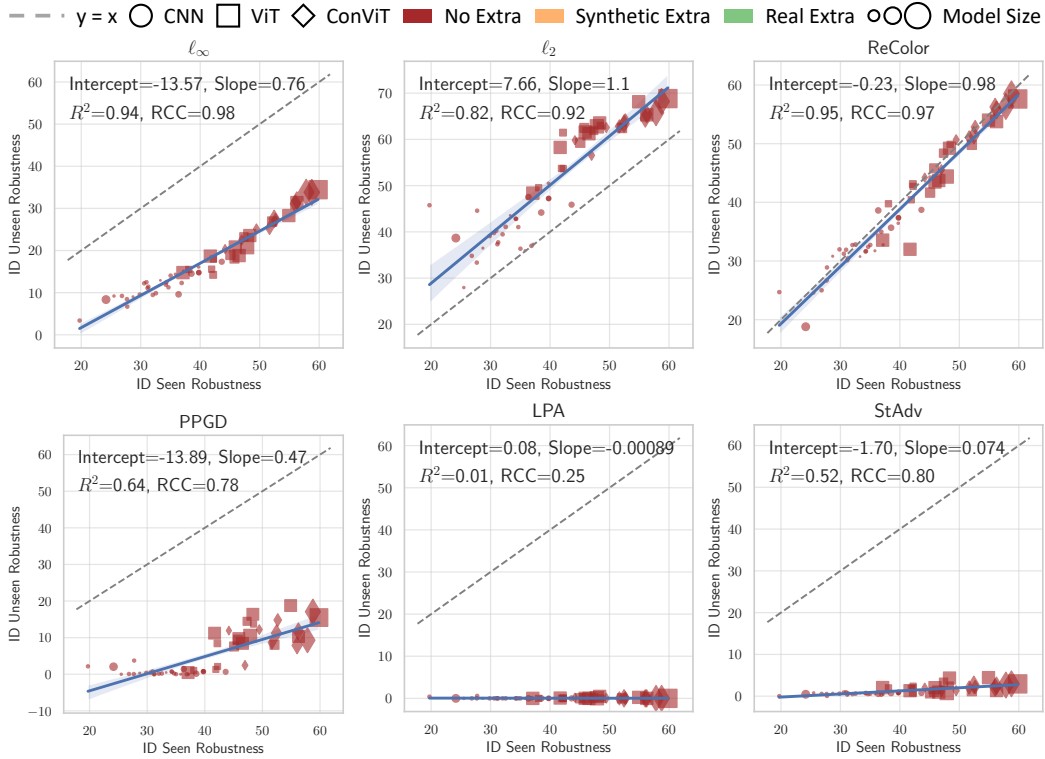

Figure 27: Correlation between seen and unforeseen robustness on ID data for ImageNet $\ell_\infty$ AT models

