# OpenReview forum: "OODRobustBench: benchmarking and analyzing adversarial robustness under distribution shift"
_ICLR.cc/2024/Conference — Submitted to ICLR 2024_

### Official Review · Reviewer_YQdD · 2023-10-30

**Soundness:** 3 good
**Presentation:** 3 good
**Contribution:** 2 fair
**Rating:** 6
**Confidence:** 3

**Summary:**

This paper provides a large-scale benchmark for evaluating the adversarial robustness of models on datasets under distribution shift (OOD robustness). It supports 23 dataset-wise shifts (e.g., image corruptions) and 6 threat-wise shifts (i.e., different adversarial attacks) and is used to assess the OOD robustness of 706 pre-trained models. Based on the experimental analysis, this work has some insightful findings, e.g.,  1) robustness degrades significantly under distribution shift, 2) ID accuracy (robustness) strongly correlates with OOD accuracy (robustness) in a linear relationship. Based on the finding of the linear relationship, the authors propose to predict the OOD performance of models using ID performance. Finally, some explorative studies such as data augmentation have been conducted and demonstrated to be useful to improve the OOD robustness.

**Strengths:**

- Large scale evaluation with diverse distribution shift. OODRobustBench supports 29 types of distribution shifts and 706 types of models that can provide a good platform for researchers to further analyze the OOD robustness problem.

- Some interesting findings, e.g., adversarial training can boost the correlation between the ID accuracy (robustness) and OOD accuracy (robustness), and no evident correlation when ID and OOD metrics misalign.

- Useful guidance. I like the part of Section 5 that explores the usefulness of using multiple methods (even though they are from existing works) to enhance the OOD robustness.

- This benchmark systematizes the question of whether robustness acquired againt a specific threat model transfers to other threat models. Although this is not the first work that investigate this aspect, this is to my knowledge the first large-scale benchmark considering this question.

**Weaknesses:**

- Only one seen attack method (MM5) has been used for the evaluation, which is not practical for robustness analysis.
Some findings are expected or have already been studied in existing works.

- Some works [1, 2, 3] studied the linear correlation between ID performance and OOD performance, the authors need to add some related discussions. Instead of the finding of the correlation between ID accuracy (robustness) and OOD accuracy  (robustness) that is expected, the finding that there is a weak correlation between ID accuracy and OOD robustness for ImageNet is more attractive and needs a more detailed explanation.

- Section 4.3 analyzes the upper limit of OOD performance using the linear correlation. However, the authors did not consider factors that could further improve the limit such as OOD generalization methods which makes the conclusion not that convincing. Instead of analyzing the limits,  it is better to try more model accuracy prediction methods such as [1, 2, 3] to evaluate their effectiveness in assessing OOD robustness.

[1] Agreement-on-the-Line: Predicting the Performance of Neural Networks under Distribution Shift, Neurips 2022.
[2] Leveraging Unlabeled Data to Predict Out-of-Distribution Performance, ICLR 2022.
[3] Are labels always necessary for classifier accuracy evaluation? CVPR 2021

This is a borderline paper. Even though this paper provides the first benchmark for OOD robustness evaluation, there are some concerns that need to be addressed, the limited seen attack methods used, some findings already revealed by existing works, and lack of the study of OOD generalization methods.

**Questions:**

1. Compared to existing works [1, 2, 3], can you please summarize the new findings from OODRobustBench?
2. After improving the OOD robustness using the methods in Section 5, do you think the findings revealed by the previous sections will change?
3. Do you think other OOD generalization methods like unsupervised representation learning for OOD generalization [4] can help increase the limit of OOD robustness?

[1] Agreement-on-the-Line: Predicting the Performance of Neural Networks under Distribution Shift, Neurips 2022.
[2] Leveraging Unlabeled Data to Predict Out-of-Distribution Performance, ICLR 2022.
[3] Are labels always necessary for classifier accuracy evaluation? CVPR 2021
[4] Towards Out-Of-Distribution Generalization: A Survey. Arxiv

---

> ### Author Response · Authors · 2023-11-20
>
> > Only one seen attack method (MM5) has been used for the evaluation, which is not practical for robustness analysis.
>
> As mentioned in Appendix B.2.1, we use the MM5 attack [5] because of its strong trade-off between efficiency and effectiveness. MM5 has about 0.5% lower attack success rate than the commonly used AutoAttack [6], but it is **32× faster** (discussed in Appendix B.2.1). This means MM5 is a much more practical choice, especially for our large-scale analysis of hundreds of models in OODRobustBench. Please feel free to clarify if this does not address your question.
>
> [5] Gao et al., *Fast and Reliable Evaluation of Adversarial Robustness with Minimum-Margin Attack*, ICML 2022.
>
> [6] Croce and Hein, *Reliable Evaluation of Adversarial Robustness with an Ensemble of Diverse Parameter-free Attacks*, ICML 2020.
>
> > Some works [1, 2, 3] studied the linear correlation between ID performance and OOD performance, the authors need to add some related discussions.
>
> [1,2,3] focus on predicting the OOD accuracy of non-robust models when OOD data is *unlabeled*, while our work predicts the OOD adversarial robustness of robust models based on labeled OOD data. We quickly summarize these works below:
>
> - [1] predicts OOD accuracy based on the linear trend of the prediction agreement of two pairs of models.
> - [2] learns a threshold for prediction score on labeled ID data and predicts OOD accuracy using the fraction of OOD examples whose prediction score is above the threshold.
> - [3] predicts OOD accuracy based on a linear trend, learned on the synthetic datasets, between accuracy and distribution shift (measured by Fréchet distance).
>
> The linear trend of adversarial robustness discovered in our work suggests that these methods are potentially applicable to predicting OOD adversarial robustness. We summarize our new findings compared to this line of work in [shared response](https://openreview.net/forum?id=RnYd44LR2v&noteId=mJy1d8sGtg). We have added this discussion to our paper in Appendix A.
>
> > …the finding that there is a weak correlation between ID accuracy and OOD robustness for ImageNet is more attractive and needs a more detailed explanation
> >
>
> There is no correlation between ID accuracy and OOD robustness for CIFAR-10 but a weak correlation for ImageNet because of the *inferior* models on CIFAR-10 which have high accuracy yet poor robustness. These models cause OOD robustness to not consistently increase with the ID accuracy. These inferior models are mainly produced by some of our custom training receipts and take a considerable proportion of our CIFAR-10 model zoo, whereas the model zoo of ImageNet is dominated by ones from public sources. We have added this discussion in section 4.1 in our paper.
>
> > Section 4.3 analyzes the upper limit of OOD performance using the linear correlation. However, the authors did not consider factors that could further improve the limit such as OOD generalization methods which makes the conclusion not that convincing.
> >
>
> We note that our goal is to benchmark and analyze the OOD adversarial robustness of existing robust models. Thus, **the drawn conclusion on linear trend and upper limit mainly applies to existing AT methods and enhancement techniques such as extra data, model scaling, etc.** Since OOD generalization methods, such as, IRM [7] and REx [8], are rarely used in the existing AT methods, their robustness to attacks is unclear and so not reflected in our conclusions. In fact, one of the implications of this work is to hopefully spur the future development of new approaches that can cope with distribution shifts better and achieve OOD robustness beyond our prediction. We have added this discussion to our paper in Appendix H.1.
>
> [7] Arjovsky et al. Invariant Risk Minimization. arxiv 2020.
>
> [8] Kruger et al. Out-of-Distribution Generalization via Risk Extrapolation. ICML. 2021.
>
> > Instead of analyzing the limits, it is better to try more model accuracy prediction methods such as [1, 2, 3] to evaluate their effectiveness in assessing OOD robustness.
>
> The core of [1, 2, 3] is to predict OOD accuracy without relying on the label of OOD data. However, it is challenging to apply them to predict OOD adversarial robustness in the same manner because the generation of strong OOD adversarial examples typically requires labels. Supposing we have already generated these adversarial examples, we run the experiments for CIFAR-10 models with CIFAR-10.1 and Impulse noise shifts and find that a linear trend is also observed in the agreement between the predictions of any two pairs of robust models: R2 is 0.95 for Impulse noise shift and 0.99 for CIFAR-10.1 shift (see Figure 18 for the plots). This suggests that the method in [1] is also effective in predicting OOD adversarial robustness. Due to the time limit, we can only experiment [1] and have to postpone [2,3]. We have added this new result and discussion to our paper in Appendix G.

---

> ### Author Response · Authors · 2023-11-20
>
> > After improving the OOD robustness using the methods in Section 5, do you think the findings revealed by the previous sections will change?
> >
>
> No, the overall trend would not change since the models in section 5 have been already included in the analysis in section 4.
>
> > Do you think other OOD generalization methods like unsupervised representation learning for OOD generalization [4] can help increase the limit of OOD robustness?
> >
>
> The OOD generalization methods alone are unlikely to increase the limit because they offer little or no adversarial robustness (see the OOD robustness of leading corruption defenses from RobustBench in Tab. 1 in our paper). Whether an OOD generalization method (when combined with adversarial training) can improve the OOD robustness likely differs from one method to another. On the one hand, it is observed in [9] that variance-based risk extrapolation (REx), when combined with adversarial training, improves OOD adversarial robustness under threat shifts.
>
> On the other hand, regarding unsupervised representation learning, the methods listed in [4] have never been applied before to adversarial training so it is unclear whether they will help or not. There is a line of work on self-supervised adversarial training which is close to [4]. We have evaluated one of these methods named Adversarial Contrastive Learning (ACL) [10] which combines self-supervised contrastive learning with adversarial training. The effective robustness under dataset shift and threat shift is 0.12% and 0.64%, suggesting only marginal benefit in improving the limit of OOD robustness. If the reviewer has other works of interest, we are happy to discuss and run the experiments with them if feasible. We have added this discussion to our paper in Appendix H.1.
>
> | Dataset | Threat | Training |   Model  | ID Acc. | ID Rob. | OOD_d Acc. | OOD_d Rob. | OOD_d EAcc. | OOD_d ERob. | OOD_t Rob. | OOD_t ERob. |
> |:-------:|:------------:|:--------:|:--------:|:-------:|:-------:|:----------:|:----------:|:-----------:|:-----------:|:----------:|:-----------:|
> | CIFAR10 |     Linf     |    ACL   | ResNet18 |  82.31  |  49.38  |    64.19   |    30.11   |    -0.56    |     0.12    |    28.53   |     0.64    |
>
> [9] Ibrahim et al., Towards Out-of-Distribution Adversarial Robustness, Arxiv 2023
>
> [10] Jiang et al., Robust Pre-Training by Adversarial Contrastive Learning, NeurIPS 2020

---

### Official Review · Reviewer_1Jrx · 2023-11-01

**Soundness:** 3 good
**Presentation:** 2 fair
**Contribution:** 2 fair
**Rating:** 5
**Confidence:** 4

**Summary:**

This paper studies the out-of-distribution (OOD) generalization of adversarial robustness, when there is a shift on either the test data or threat model of the adversarial perturbation. The authors built a benchmark for such evaluation and conducted a comprehensive evaluation of many models. The paper shows a linear trend between the in-distribution (ID) performance and OOD performance on many models under adversarial attacks, but there are also models showing stronger robustness beyond the linear prediction in Section 5.

**Strengths:**

* The paper presents a benchmark (OODRobustBench) for evaluating the adversarial robustness under distribution shifts (either a shift on the test data or threat model of the adversarial perturbation), based on existing OOD test sets and variants of adversarial perturbations.
* The paper conducted a comprehensive evaluation on many models.
* The paper showed a linear trend between the ID adversarial robustness and the OOD adversarial robustness on most of the models, which is consistent with the linear trend observed in prior works (Taori et al., 2020, Miller et al., 2021) on dataset shifts without adversarial attack.

**Weaknesses:**

* This work is a straightforward combination of existing evaluations with little new contribution or understanding:
  * For the evaluation with data shifts, compared to existing works on evaluation with OOD datasets, this work simply adds existing adversarial attacks. Methods and conclusions are almost the same as previous works (Taori et al., 2020, Miller et al., 2021) on the linear trend.
  * For the evaluation on threat shifts, this work is almost the same as existing works mentioned in the "robustness against unforeseen adversarial threat models" paragraph in Section 2 but only adds more existing models.

* Some discussions on the experiments are not very accurate:
  * "Surprisingly, VR also clearly boosts effective robustness under dataset shift
even though not designed for dealing with these shifts" and "Advanced model architecture significantly boosts robustness and effective robustness under both types of shift over the classical architecture": I don't agree these "significantly boost" the effective robustness. The gains are only around 1%~2%, which are not larger than the normal variations between different models in the linear fit (Figure 1).
  * "Training with extra data boosts both robustness and effective robustness for both dataset and threat shifts compared to training schemes without extra data (see Fig. 6a). The improved effective robustness suggests that this technique induces extra OOD generalizability." It is already known that altering the training data can interfere with the traditional effective robustness evaluation rather than truly improve effective robustness (Shi et al., 2023).
  * In Section 5, the authors rename the existing effective robustness from previous works (which have been widely adopted) into "effective accuracy" while redefine "effective robustness" to be the effective robustness under adversarial attacks. This is confusing. I would suggest the authors keep the original definition for effective robustness but give a new name for the particular effective robustness in this work (e.g, adversarial effective robustness).

Shi, Z., Carlini, N., Balashankar, A., Schmidt, L., Hsieh, C. J., Beutel, A., & Qin, Y. (2023). Effective Robustness against Natural Distribution Shifts for Models with Different Training Data. arXiv preprint arXiv:2302.01381.

**Questions:**

* How are the results and conclusions in this paper fundamentally different from those in existing works? (See the weaknesses above.) What are the new implications of this work, beyond those already known in existing works (vulnerability against distribution shifts, linear trend, etc.)?

---

> ### Author Response · Authors · 2023-11-20
>
> > This work is a straightforwad combination of existing OOD datases and adversarial attacks
>
> Technically yes. Our work focuses on standardizing the evaluation of OOD adversarial robustness and understanding this generalization, instead of proposing new OOD datasets and attacks. Building benchmarks based on the existing datasets and attacks is the same as the construction of other public benchmarks like RobustBench [1], RobustART [2], and MultiRobustBench [3]. Please also refer to [shared response](https://openreview.net/forum?id=RnYd44LR2v&noteId=mJy1d8sGtg) for a summary of our contributions.
>
> Reference:
>
> [1] croce et al., RobustBench: a standardized adversarial robustness benchmark, NeurIPS 2021
>
> [2] Tang et al., RobustART: Benchmarking Robustness on Architecture Design and Training Techniques, T-PAMI 2022
>
> [3] Dai et al., MultiRobustBench: Benchmarking Robustness Against Multiple Attacks, ICML 2023
>
> > For the evaluation with data shifts, compared to existing works on evaluation with OOD datasets, this work simply adds existing adversarial attacks. Methods and conclusions are almost the same as previous works (Taori et al., 2020, Miller et al., 2021) on the linear trend.
>
> Please see [shared response](https://openreview.net/forum?id=RnYd44LR2v&noteId=mJy1d8sGtg) on the top.
>
> > For the evaluation on threat shifts, this work is almost the same as existing works mentioned in the "robustness against unforeseen adversarial threat models" paragraph in Section 2 but only adds more existing models.
> >
>
> Please see [shared response](https://openreview.net/forum?id=RnYd44LR2v&noteId=mJy1d8sGtg) on the top.
>
> > How are the results and conclusions in this paper fundamentally different from those in existing works? (See the weaknesses above.) What are the new implications of this work, beyond those already known in existing works (vulnerability against distribution shifts, linear trend, etc.)?
> >
>
> Please see [shared response](https://openreview.net/forum?id=RnYd44LR2v&noteId=mJy1d8sGtg) on the top.
>
> > I don't agree these "significantly boost" the effective robustness. The gains are only around 1%~2%, which are not larger than the normal variations between different models in the linear fit (Figure 1).
> >
>
> First, when we say boost effective robustness we mean improving effective robustness over the baseline (i.e., PGD for VR and ResNet for ConvNeXt, Swin, and ViT) because this shows the benefit of switching from one method to another. **It may be less meaningful to look at the absolute value of effective robustness without comparing against the baseline** because the baseline may have a non-zero, or even negative, effective robustness. If measured so, the improvement is about 2% under OOD_d for VR and about 1.5%, 1.3%, 2.75% under OOD_d and 3%, 3%, 4.5% under OOD_t for ConvNeXt, Swin, and ViT respectively.
>
> Second, the standard deviation is 0.77% for CIFAR-10 under OOD_d (corresponding to the setting of VR) and 0.96% and 1.55% for ImageNet under OOD_d and OOD_t (corresponding to the setting of architecture). **The above improvements are all greater than one standard deviation.** In addition, we have also done a $t$-test and the $p$-value is $< 10^{-3}$ for VR under OOD_d and 0.009, 0.023, $< 10^{-3}$ under OOD_d, and 0.008, 0.008, $< 10^{-3}$ under OOD_t for ConvNeXt, Swin, and ViT respectively. **All** $p$**-values are much less than the usual significance level of 0.05 suggesting the improvement is significant**.
>
> > It is already known that altering the training data can interfere with the traditional effective robustness evaluation rather than truly improve effective robustness (Shi et al., 2023).
>
> We argue that the conclusion of Shi et al. is not applicable to our case. We quote the main conclusion from Shi et al.:  *“We observe that when there is a mismatch in the training data and the ID test data, the models appear to have greater effective robustness, as their performance on the ID test data tends to be lower”.*
>
> The mismatch in their work refers to e.g. YFCC-trained model and ImageNet test set. This is **different from training with extra data in our work,** where extra data are controlled to have a distribution as close as possible to the original training distribution so **no significant mismatch between training data and ID test data**. For example, common extra real data for CIFAR-10 are from 80 Million Tiny Images of which CIFAR-10 is a subset. Another reason that the conclusion from Shi et al. is not applicable is that the models trained with extra data have higher ID test performance instead of lower as observed by Shi et al (quoted ”as their performance on the ID test data tends to be lower”).
>
> > I would suggest the authors keep the original definition for effective robustness but give a new name for the particular effective robustness in this work (e.g, adversarial effective robustness).
> >
>
> Thanks for your suggestion. We have renamed it in the paper as suggested.

---

> > ### Comment · Reviewer_1Jrx · 2023-12-03
> > **Thanks for the response**
> >
> > Overall, I think the systematic empirical results and findings have merits for future works, but I still feel the significance of the contributions is kind of limited, compared to the previous works. I'm on the borderline for this paper.

---

### Official Review · Reviewer_ogvR · 2023-11-02

**Soundness:** 4 excellent
**Presentation:** 3 good
**Contribution:** 3 good
**Rating:** 6
**Confidence:** 5

**Summary:**

The paper studies the adversarial robustness of image classifiers in presence of out-of-distribution (OOD) tasks. Given a model which has been (adversarially) trained on a specific dataset to be robust to a chosen attack (in-distribution task), the paper suggests to test how its robustness behaves when using either images from a different distribution (OOD dataset) or a different type of attack (OOD threat model): this provides an overview of generalization of robustness. Moreover, evaluating many existing and newly trained classifiers, the paper provides insights on which are the most relevant factors to achieve OOD robustness, which might be used to develop techniques for more robust models.

**Strengths:**

- Studying the generalization of adversarial robustness is a relevant topic (for example, as at test-time attackers are not limited to use the same attack seen during training), which has received only limited attention by prior works.

- The paper provides extensive evaluations on many classifiers spanning different datasets and (seen) threat models. This gives clear trends and insights about how to improve future robust models for better generalization.

**Weaknesses:**

- Similar analyses are already present in prior works, although on a (sometimes much) smaller scale, and then the results are not particularly surprising. For example, the robustness of CIFAR-10 models on distributions shifts (CIFAR-10.1, CINIC-10, CIFAR-10-C, which are also included in this work) was studied on the initial classifiers in RobustBench (see [Croce et al. (2021)](https://arxiv.org/abs/2010.09670)), showing a similar linear correlation with ID robustness. Moreover, [A, B] have also evaluated the robustness of adversarially trained models to unseen attacks.

- A central aspect of evaluating adversarial robustness is the attacks used to measure it. In the paper, this is described with sufficient details only in the appendix. In particular for the non $\ell_p$-threat models I think it would be important to discuss the strength (e.g. number of iterations) of the attacks used, since these are not widely explored in prior works.

[A] https://arxiv.org/abs/1908.08016
[B] https://arxiv.org/abs/2105.12508

**Questions:**

- See the weaknesses mentioned above.

- Many non $\ell_p$ attacks have been proposed (see e.g. [A, B]). Is there a specific reason for the choice of those used in the paper?

---

> ### Author Response · Authors · 2023-11-20
>
> > Similar analyses are already present in prior works, although on a (sometimes much) smaller scale, and then the results are not particularly surprising. For example, the robustness of CIFAR-10 models on distributions shifts (CIFAR-10.1, CINIC-10, CIFAR-10-C, which are also included in this work) was studied on the initial classifiers in RobustBench (see [Croce et al. (2021)](https://arxiv.org/abs/2010.09670)), showing a similar linear correlation with ID robustness. Moreover, [A, B] have also evaluated the robustness of adversarially trained models to unseen attacks.
> >
>
> We acknowledge that there exist previous works that evaluate adversarial robustness under dataset shifts and threat shifts. However, as pointed out by the reviewer, the scale of these previous works is rather small. For instance, RobustBench observes linear correlation only for three shifts on CIFAR-10 based on 39 models with either ResNet or WideResNet architectures. In such a narrow setting, it is actually neither surprising to see a linear trend nor reliable for predicting OOD performance. By contrast, our conclusion is derived from much more shifts on CIFAR-10 and ImageNet based on 706 models. Importantly, our model zoo covers a diverse set of architectures, robust training methods, data augmentation techniques, and training set-ups. This makes our conclusion more generalizable and the observed (almost perfect) linear trend much more significant.
>
> Similarly, [A] and [B] only test a few models under threat shifts. Those methods are usually just the baseline AT method plus different architectures or the relevant defenses, e.g., jointly trained with multiple threats. It is unclear how the state-of-the-art robust models perform under threat shifts. By conducting a large-scale analysis, we find that **those SOTA models generalize poorly to other threats while also discovering several methods that have relatively inferior ID performance but superior OOD robustness under threat shift.** Our analysis therefore facilitates future works in this direction by identifying what techniques are ineffective and what are promising.
>
> We have added this discussion in appendix A, as well as the reference to [B], to our paper.
>
> Please also refer to [shared response](https://openreview.net/forum?id=RnYd44LR2v&noteId=mJy1d8sGtg) for a summary of our new findings compared to the previous works.
>
> > A central aspect of evaluating adversarial robustness is the attacks used to measure it. In the paper, this is described with sufficient details only in the appendix. In particular for the non-$\ell_p$ threat models I think it would be important to discuss the strength (e.g. number of iterations) of the attacks used, since these are not widely explored in prior works.
> >
>
> Thanks for your suggestion. The complete specification of used attacks was given in Appendix A.2. We have moved this discussion to the main text in section 3.1.
>
> > Many non-$\ell_p$ attacks have been proposed (see e.g. [A, B]). Is there a specific reason for the choice of those used in the paper?
> >
>
> Ideally, we would like to include more non-Lp attacks to understand the generalization of adversarial robustness more thoroughly. However, adding attacks will cause substantial computational costs because of our large model zoo. We choose the current ones because they are sufficiently effective, cover perceptible (ReColor, StAdv) and imperceptible (PPGD, LPA) attacks, and have been widely used (each of them is cited 100+). Nevertheless, we are happy to include other non-Lp attacks if any of your interest. We have added this discussion to our paper in Appendix B.2.2.

---

> > ### Comment · Reviewer_ogvR · 2023-11-22
> >
> > I thank the authors for the response and updates of the paper.
> >
> > I think that, although most of the conclusion of the paper are in line with what already suggested by prior works, the extensive and careful evaluation has value and provides some useful insights. Then I keep my original score.

---

> > > ### Author Response · Authors · 2023-11-22
> > >
> > > Many thanks for your reply and recognizing our work. Please let us know if anything else we could help.

---

### Official Review · Reviewer_UQFk · 2023-11-09

**Soundness:** 4 excellent
**Presentation:** 3 good
**Contribution:** 2 fair
**Rating:** 6
**Confidence:** 5

**Summary:**

The proposed approach combines the research direction of adversarial robustness and domain shift into a single benchmark: OODRobustBench. It measures how the adversarial robustness of networks trained on in-distribution data varies when evaluated on test data under distribution shift. It provides an evaluation over 706 robust models to draw insights into correlation of OOD and in-distribution (ID) robustness, upper limit on OOD robustness, and effect of training setup on OOD robustness.

**Strengths:**

This paper thoroughly explores the intersection of adversarial robustness and OOD generalization works in developing the OODRobustBench benchmark. It considers multiple ablations across ID and OOD robustness and performs 60.7K adversarial evaluations.The paper is also well written and very easy to parse.

**Weaknesses:**

While the proposed evaluation is rigorous, I believe that approach fall-short of being a standardized benchmark. It’s unclear what criterions are allowed for robust models. Would the benchmark include all adversarial robust approaches, such as preprocessing based defenses. If yes, how would the trend with robustness in such approaches correlate with OOD robustness?

Is the specific trend in natural accuracy between IN and OOD data particular to robust models? Can authors provide some results/citation on how the correlation between IN-OOD accuracy correlated with IN-ODD accuracy on non-robust models?

**Questions:**

In figure 1.4, interestingly ID unseen robustness is higher than the seen robustness at lower robustness levels. This result is apparently counter-intuitive, as the trend quickly diminishes, at higher robustness levels. Can authors shed more light on this phenomenon.

Can authors also provide a concrete comparison on how the proposed benchmark is different from robustbench, in particular in-terms of benchmarking in-distribution robustness.

To aggregate performance across different attacks in each group (OOD_d/OOD_t), shouldn’t harmonic mean be used to achieve better stability.

---

> ### Author Response · Authors · 2023-11-20
>
> > It’s unclear what criterions are allowed for robust models. Would the benchmark include all adversarial robust approaches, such as preprocessing based defenses.
>
> We follow the same criteria as the popular benchmarks (RobustBench, MultiRobustBench, etc), which only include robust models that (1) have in general non-zero gradients w.r.t. the inputs, (2) have a fully deterministic forward pass (i.e. no randomness) and (3) do not have an optimization loop. These criteria include most AT models, while **excluding most preprocessing methods** because they rely on randomness like [1] or inner optimization loop like [2] which leads to false security, i.e., high robustness to the nonadaptive attack but vulnerable to the adaptive attack.
>
> Meanwhile, we acknowledge that evaluating dynamic preprocessing-based defenses is still an active area of research. It is tricky [3], and there has not been a consensus on how to evaluate them. So now, we exclude them for a more reliable evaluation. We will keep maintaining this benchmark, and we would be happy to include them in the future if the community has reached a consensus on that (e.g., if these models are merged into RobustBench). We have added this discussion to our paper in Appendix C.1.
>
> > how the correlation between IN-OOD accuracy correlated with IN-ODD accuracy on non-robust models?
>
> A similar trend between ID and OOD accuracy is also observed previously in [4] for non-robust models. Robust models exhibit a stronger linear correlation between ID and OOD accuracy for most corruption shifts (Fig. 3 in the paper). Particularly, the boost on linearity is dramatic for shifts including Impulse, Shot and Gaussian noises, Glass blur, Pixelate, and JPEG. For instance, R2 surges from 0 (no linear correlation) for non-robust models to 0.84 (evident linear correlation) for robust models with Gaussian noise data shifts. This suggests that, for robust models, predicting OOD accuracy from ID accuracy is more faithful and applicable to more shifts. We have added this discussion to our paper in Appendix A.
>
> > In figure 1.4, interestingly ID unseen robustness is higher than the seen robustness at lower robustness levels.
>
> This is indeed interesting. Figure 1.4 depicts the trend of Lp threat shift that consists of Linf with an epsilon of 12/255 (higher than the training epsilon of 8/255) and L2 with an epsilon of 0.5. The phenomenon results from the fact that L2-0.5 (unseen) robustness is much higher than Linf-8/255 (seen) robustness for lower-robustness models (see Figure 25 in the appendix). It seems that lower-robustness models have a better generalization ability to the unseen threat L2. However, such ability degenerates for models at higher robustness levels. Consequently, the slope of the linear trend in Figure 1.4 is lower than 1 depicting the cross of two trends in the figure. Last, we note that such remarkable generalization ability to unseen L2 threat is also observed on both low- and high-robustness ImageNet Linf-trained models (see Figure 27 in the appendix).
>
> > a concrete comparison on how the proposed benchmark is different from robustbench.
>
> Our benchmark focuses on OOD adversarial robustness while RobustBench focuses on ID adversarial robustness. Specifically, our benchmark contrasts RobustBench in the datasets and the attacks. We use CIFAR-10.1, CIFAR-10.2, CINIC, and CIFAR-10-R (ImageNet-V2, ImagetNet-A, ImageNet-R, ObjectNet) to simulate input data distribution shift for the source datasets CIFAR-10 (ImageNet), while RobustBench only uses the latter source datasets. We use PPGD, LPA, ReColor, StAdv, Linf-12/255, L2-0.5 (PPGD, LPA, ReColor, StAdv, Linf-8/255, L2-1) to simulate threat shift for the training threats Linf-8/255 (L2-0.5), while RobustBench only evaluates the same threats as the training ones. We have added this discussion to our paper in Appendix A.
>
> > To aggregate performance across different attacks in each group (OOD_d/OOD_t), shouldn’t harmonic mean be used to achieve better stability.
>
> Thanks for your suggestion. We think harmonic mean may be not applicable to aggregate our data. First, harmonic mean tends to emphasize the smaller values in the data, i.e., more severe shifts. This may not reflect the true importance of each shift. Second, the calculation of harmonic mean involves the reciprocal of each data point, which means harmonic mean is not applicable to the data with zeros. It is possible for the robustness, especially under non-Lp threat shifts, to be zeros.
>
> Reference:
>
> [1] Guo et al., *Countering Adversarial Images using Input Transformations*, ICLR 2018.
>
> [2] Samangouei et al., *Defense-GAN: Protecting Classifiers Against Adversarial Attacks Using Generative Models*, ICLR 2018.
>
> [3] Croce et al., *Evaluating the Adversarial Robustness of Adaptive Test-time Defenses*, ICML 2022.
>
> [4] Miller et al., *Accuracy on the Line: on the Strong Correlation Between Out-of-Distribution and In-Distribution Generalization*, ICML 2021.

---

### Author Response · Authors · 2023-11-20
**Shared response**

We would like to thank the reviewers for taking the time to review our paper and for all the constructive feedback. The paper has been revised (marked blue) according to the comments. Please find our response to the common questions below.

> Contributions of our work
>

The value of OODRobustBench lies in its comprehensiveness and the scientific contributions we and the community can draw from it. We re-emphasize our contributions below:

1. OODRobustBench is the first systematic evaluation of OOD adversarial robustness.
2. This benchmark encompasses 29 types of shifts and 706 pre-trained models, most of which are not in RobustBench, into a single open-source platform. We use it to conduct a large-scale study to understand the current state of the OOD adversarial robustness problem.
3. Multiple crucial findings are discovered: (i) the lack of OOD robustness of existing models, (ii) the linear correlation between ID and OOD robustness, and (iii) predicted trends on OOD adversarial robustness.

> New findings on the current state of OOD adversarial robustness
>

The models studied before for OOD robustness are usually naive and quite limited. Lots of SOTA models are not covered, making the current state of OOD adversarial robustness unclear. Through our large-scale study, we discover novel and useful findings (see section 3.2) about OOD generalization of adversarial robustness:

1. SOTA robust models, e.g. the top 10 models from RobustBench CIFAR-10 Linf leaderboard, lose 25-30% of their robustness under distribution shifts.
2. **The higher the ID robustness of the model, the more robustness degrades under the shifts.** This suggests that while great progress has been made in improving ID robustness, we only gain diminishing returns under the distribution shifts
3. **Several Linf-trained models with inferior ID robustness but superior OOD robustness under *threat shifts* esp. against non-Lp attacks**. This challenges a common belief that Lp-trained models generalize poorly to non-Lp threats.
4. Some methods suffer from **abnormally catastrophic degradation** (e.g. drop by about 40 points) in robustness under noise shifts. This unexpected result can be dangerous in the real world if goes unnoticed.
5. A complete landscape about the sensitivity of existing models to a diverse set of distribution shifts. Shifts like CIFAR-10-R, CIFAR-10.2, and all ImageNet shifts are never evaluated before for OOD robustness.

> New findings compared to linear trend of accuracy for non-robust models.
>
1. **Compared to non-robust models, robust models generally exhibit a stronger linear correlation between ID and OOD accuracy** (section 4.1). For some shifts, the trend changes from no or weak correlation for non-robust models to a very strong one for robust models.
2. **The discovery of linear trend for adversarial robustness** (sections 4.1 and 4.2). (argued below why this is new and not expected)
3. **The linear trend of adversarial robustness is stronger** (section 4.1) **but has a much lower slope** (section 4.3). This limits the predicted upper bound of the OOD adversarial robustness, suggesting either an inherent limitation or a need for a new approach to improving OOD adversarial robustness.
4. **No evident correlation when ID and OOD metrics misalign**, but weak correlation for ImageNet (section 4.1). This suggests that it is unreliable to predict OOD accuracy (robustness) using ID robustness (accuracy).
5. A connection between dataset shift and threat shift: surprisingly, **the unforeseen (threat shifts) robustness correlates even more strongly with OOD adversarial robustness than ID seen robustness** (section 4.2).

> Is the linear trend of robustness really expected given the linear trend of accuracy?
>

We do not believe so. There is a well-known trade-off between accuracy and robustness in the ID setting. We further confirm this fact for the models we evaluate in Fig. 11 in the appendix. This means that accuracy and robustness usually go in opposite directions making the linear trend we discover in both particularly interesting. Furthermore, the threat shifts as a scenario of OOD are unique to adversarial evaluation and were thus never explored in the previous studies of accuracy trends.

---

### Meta-Review · Area_Chair_X5du · 2023-12-11

**Metareview:**

The authors propose a benchmark/analysis of adversarially robust models under distribution shift (e.g. adding corruptions and then testing adv. robustness) or threat shift (testing under a different threat model then it was trained on). They present several findings how ID and OOD performance are related. All reviewers appreciate the large-scale analysis of the paper (in total >700 models) but question that the findings are particularly novel or surprising. There were also concerns about the reliabilty of the results given the employed attacks.

Strengths:
- large scale analysis using a large number of models from RobustBench plus a large number of specifically trained models
- show relationships of ID and OOD robustness (even though this has been analyzed on smaller scale before as mentioned by the reviewers)
- the novelty of the findings seems a bit limited given prior work but given the large-scale analysis the statistics are much better

Weaknesses:
- the evaluation of non-$\ell_p$-attacks seems weak (e.g. only 40 iterations for PPGD). It is unclear, how reliable the evaluation is which is a critical point for an adversarial robustness paper
- the evaluation of $\ell_p$-attacks is done with MM5. While the authors have compared this to AutoAttack no statistics are provided. Apparently, larger deviations on two models have been observed (numbers are not provided). The author state that they then used MM+ and the numbers were similar to AutoAttack. But that should imply that they always have to use always MM+ (or AutoAttack) as otherwise the robustness evalution is not reliable. One downside of MM+ is that it includes no black-box attack (it is basically similar to the APGD parts of AutoAttack) and thus it is in my point of view not suited for a benchmark evaluation of adversarial robustness.

Post-rebuttal some reviewers expressed concerns about the robustness evaluation. Given that the authors envision this work as a benchmark and thus this has a potential guiding effect for the community, the reliability of the evaluation is crucial and thus the authors have to provide more evidence how they ensure this in practice. Thus while I appreciate very much the large scale effort of this paper, the authors have to ensure that their robustness evaluations are reliable.

**Justification For Why Not Higher Score:**

Doubts about the adversarial robustness evaluation.

**Justification For Why Not Lower Score:**

N/A

---

### Decision · Program_Chairs · 2024-01-16

Reject